# Development of visual motion integration involves coordination of multiple cortical stages

**Augusto A Lempel[1,2][†], Kristina J Nielsen[1,2]***

[1]Solomon H. Snyder Department of Neuroscience, Johns Hopkins University School of Medicine, Baltimore, United States; [2]Zanvyl Krieger Mind/Brain Institute, Johns Hopkins University, Baltimore, United States

**Abstract** A central feature of cortical function is hierarchical processing of information. Little is currently known about how cortical processing cascades develop. Here, we investigate the joint development of two nodes of the ferret's visual motion pathway, primary visual cortex (V1), and higher-level area PSS. In adult animals, motion processing transitions from local to global computations between these areas. We now show that PSS global motion signals emerge a week after the development of V1 and PSS direction selectivity. Crucially, V1 responses to more complex motion stimuli change in parallel, in a manner consistent with supporting increased PSS motion integration. At the same time, these V1 responses depend on feedback from PSS. Our findings suggest that development does not just proceed in parallel in different visual areas, it is coordinated across network nodes. This has important implications for understanding how visual experience and developmental disorders can influence the developing visual system.

**\*For correspondence:**
kristina.nielsen@jhmi.edu

**Present address:** [†]Max Planck Florida Institute for Neuroscience, Jupiter, United States

**Competing interests:** The authors declare that no competing interests exist.

## Introduction

Information processing in the brain involves a chain of transformations across areas, linked via feed-forward and feedback loops. Little is currently known about how this interconnected network of areas develops. Yet, without a precise determination of at least the relative developmental time courses of different areas, models for cortical development remain lacking, as does our understanding of whether and how interactions between areas might shape the emerging network. The motion pathway in visual cortex provides a unique opportunity to address these questions: Its central nodes are well established in the primate, as are key pathway functions (*Born and Bradley, 2005*; *Orban, 2008*). Accordingly, there are well-characterized stimulus sets for probing these functions and a range of computational models for the motion pathway (*Baker and Bair, 2016*; *Beck and Neumann, 2011*; *Rust et al., 2006*; *Simoncelli and Heeger, 1998*; *Zaharia et al., 2019*). This holds true in particular for motion integration: Between primary visual cortex (V1) and area MT, the next motion pathway stage, the encoding of motion information transitions from extracting local edge signals to computing global motion signals (*Born and Bradley, 2005*; *Orban, 2008*; *Movshon et al., 1985*). Typically, this is demonstrated using drifting plaids, constructed from two component gratings moving in different directions (*Movshon et al., 1985*). Perceptually, plaids appear to move in a third, intermediate (pattern) direction (*Adelson and Movshon, 1982*; *Stoner et al., 1990*). By probing whether neural responses are consistent with the individual component directions or the integrated pattern direction, the degree of motion integration in a neuron or area can be assessed. Applied to V1, these stimuli reveal mostly component-driven responses (*Movshon et al., 1985*). In primate MT, on the other hand, a quarter of the population responds to the pattern direction, indicative of an increased degree of motion integration as necessary for global motion computations (*Movshon et al., 1985*). Finally, a number of models have been developed to

capture the transformation of motion signals between V1 and MT (*Baker and Bair, 2016*; *Beck and Neumann, 2011*; *Rust et al., 2006*; *Simoncelli and Heeger, 1998*; *Zaharia et al., 2019*). Thus, the stage is set – both from an experimental and computational perspective – for using the motion pathway, and specifically motion integration, to investigate the joint development of interconnected cortical areas.

To date, the visual motion pathway has been most thoroughly investigated in adult non-human primates. Much less is known about its development (*Chino et al., 1997*; *Kiorpes and Movshon, 2014*). We have recently demonstrated that area PSS in ferret visual cortex shares many similarities with primate MT (*Lempel and Nielsen, 2019*). Crucially, the same transition from local to global motion processing occurs between ferret V1 and PSS as between primate V1 and MT. At the same time, the ferret's early parturition (*Sharma and Sur, 2014*) allows easy access to the development of these stages. Here, we leveraged the strength of the ferret as a developmental animal model, combined with our previous research on PSS, to probe the development of motion integration in the motion pathway. Using single-cell recordings, we found that PSS motion integration develops after postnatal day (P) 40, during the second week after eye opening, and after the development of direction selectivity. Maturation of motion integration was reflected not only in an increasing prevalence of pattern responses, but also in how responses depended on the exact plaid configuration. We also provide evidence that visual experience likely plays a role in the development of this process.

Intriguingly, we additionally observed developmental changes in V1 responses to plaids, highlighting the need to consider multiple pathway stages simultaneously during development. V1 development included a temporary increase in motion integration and plaid response strength between P44 and P47. Inactivation of PSS demonstrated that both of these changes depended on feedback from PSS. At the same time, we expect that changes in V1 responses are likely to propagate to PSS. To test how these V1 changes might complement PSS changes during motion integration development, we implemented a two-stage model of the ferret's motion pathway. The model was then used to identify the relative contributions of V1 and PSS development to the increase in PSS pattern responses after P40. This modeling exercise suggested development of strong inhibition between preferred and null direction, internal to PSS, as an important contributor to increased motion integration. At the same time, the temporary changes in V1 plaid responses complemented PSS-internal changes to overcome possibly detrimental consequences of the increased inhibition levels, namely overall reductions in PSS pattern responses. In summary, our results reveal that development of the motion pathway involves coordination between multiple network nodes in a highly dynamic fashion.

## Results

### Developmental timeline of PSS motion integration

Using coherent plaids, we have previously demonstrated signatures of motion integration in PSS of adult ferrets (*Figure 1A*). Specifically, a group of PSS neurons responds to the pattern direction of the plaids (so called 'pattern neurons'), rather than the motion of the individual components (as would be the case for so called 'component neurons'). Here, we first aimed to establish the developmental timeline for PSS pattern responses. To this end, we used tetrodes or multi-channel silicon probes to record PSS responses in anesthetized ferrets at various ages. Previous research in primates suggests that motion integration develops after maturation of direction selectivity (*Chino et al., 1997*; *Kiorpes and Movshon, 2014*). In addition, a minimal level of direction selectivity is required for a meaningful analysis of responses to plaids. We therefore restricted our experiments to an age range after V1 direction selectivity development, which in ferrets occurs around P37 (ferrets open their eyes at about P30). Our data confirm that PSS direction selectivity is also mature at this age (see *Figure 1—figure supplement 1*).

PSS motion integration was assessed at different ages by probing neurons with plaids and gratings. Since plaids are constructed from two component gratings moving in different directions, each plaid can be described by two parameters (*Figure 1B*) – the difference in direction between the two components (dOri) and the resulting plaid direction (which bisects the two component directions). Importantly, the perception of a coherently moving plaid is maintained over a large range of dOri values (*Adelson and Movshon, 1982*). To efficiently sample a large number of dOri values and

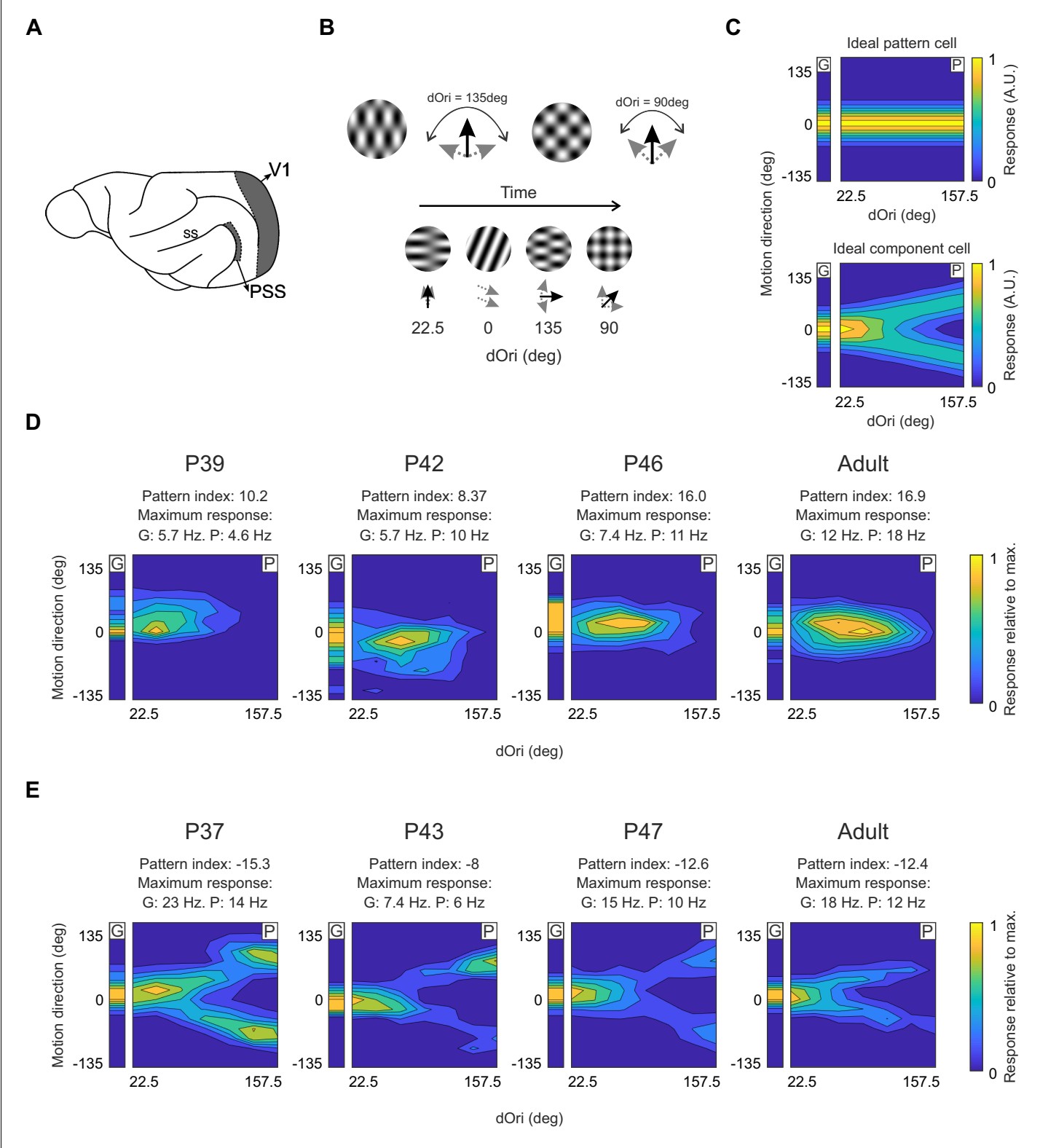

**Figure 1.** Pattern and component cells are found in PSS across all ages tested. (**A**) Sagittal view of the ferret brain indicating the location of PSS and V1 (ss: suprasylvian sulcus). (**B**) Top: Plaid stimuli with different angles between the component directions (dOri). Bottom: Illustration of the streaming stimulus paradigm: Plaids of different dOri values and directions were presented along with gratings moving in different directions (see Materials and methods). (**C**) Simulated responses of an ideal pattern (top) and component cell (bottom) to gratings and plaids. In this and the following figures, the small contour plot for each cell (labeled 'G') represents the responses to gratings only. The large 2D contour plot (labeled 'P') summarizes

*Figure 1 continued on next page*

*Figure 1 continued*

responses to all plaids as a function of dOri and direction. In both plots, direction is relative to the cell's preferred grating direction. For 'G', direction refers to the grating direction; for 'P', direction refers to the pattern direction (i.e. the perceived direction of the plaid). 'G' and 'P' are independently normalized by their respective maximum to help with visualization of the tuning properties. An ideal pattern cell would respond maximally to plaid stimuli moving in its preferred direction. An ideal component cell would respond maximally whenever one plaid component moves in the preferred direction, resulting in a response profile with a characteristic 'V'-shape. For the ideal cells, resposes were scaled so that the area under the direction curve remained constant across dOri values. As a consequence, the responses for the ideal component cell change with dOri (because different number of directions evoke strong responses). All other dOri dependencies were ignored in this illustration. (D) Responses of example PSS neurons classified as pattern cells (see Materials and methods for classification procedure) in ferrets of different ages. For each neuron, we list the pattern index, as well as the maximum response rate evoked by gratings (G) and plaids (P) (otherwise same format as in (C)). (E) Responses of example PSS component cells in ferrets of different age (same format as (D)).

The online version of this article includes the following figure supplement(s) for figure 1:

**Figure supplement 1.** PSS direction selectivity is mature at P37.

plaid directions, we presented stimuli in a streaming stimulus paradigm, in which stimuli were shown in rapid succession (stimulus update rate: 3 Hz; *Figure 1B*). We have previously shown that the results of this approach generally agree with those based on showing single stimuli more slowly, but that the sampling of a larger stimulus space noticeably improves the discriminability of pattern and component responses, in addition to generating results that are independent of a particular dOri choice (see *Rust et al., 2006*; *Smith et al., 2005* for a similar presentation method).

For every neuron probed with the streaming stimulus paradigm, we then determined its capacity to integrate local motion signals (see *Lempel and Nielsen, 2019* and Materials and methods for details). Briefly, for every neuron, we generated two predictions from the measured stimulus responses, one for pattern and one for component responses. Each prediction summarized tuning as a two-dimensional (2D) function of both the (integrated) plaid direction and dOri. Since coherent plaid motion is perceived independent of dOri, the pattern prediction assumed that the tuning peak would always occur for plaids moving in the preferred direction, regardless of dOri value (but allowing changes in peak height in a dOri-dependent fashion). Consequently, tuning profiles for pattern predictions had a single ridge along the preferred direction (*Figure 1C*, top). The component prediction, on the other hand, assumed that the individual component directions determined responses, not the integrated plaid direction. For each dOri, the strongest responses therefore should occur whenever one of the components moved in the preferred direction, resulting in a characteristic 'V'-shape in the chosen 2D tuning plot format (*Figure 1C*, bottom). As for the pattern prediction, dOri-dependent changes in maximum rates were included in the component prediction. We then compared the two predictions to the measured tuning profile of each neuron by computing Z-corrected partial correlations, resulting in a pattern correlation ($Z_P$) and a component correlation ($Z_C$). We also computed a pattern index for every neuron as $Z_P$–$Z_C$. Based on established criteria (*Smith et al., 2005*), neurons were classified as pattern neurons if $Z_P$ was significantly higher than $Z_C$ (using $p<0.1$); component neurons met the opposite criterion (see *Figure 2C* and Materials and methods for precise definitions of category boundaries).

Across all ages tested (P37–13 months), we were able to observe pattern and component cells in PSS (*Figure 1D,E*). Nonetheless, the degree of PSS motion integration underwent pronounced changes with age. As a first step in assessing the time course of motion integration development, we computed the median pattern index for every animal. *Figure 2A* shows these medians as a function of age (Note that for some animals less than three neurons were recorded. We included these animals for completeness but indicate the corresponding data points in *Figure 2A*.) While this plot confirmed that a range of pattern indices could be observed at every age, it also showed that the lowest median pattern indices generally occurred in animals before P41. After P41, the range of median pattern indices across animals of the same age was generally similar to that of adults. We therefore divided all developmental data into two groups, P37–40 and P41–47. A third group consisted of adult animals (age range P100–448). As expected, the per-animal pattern indices in the P37–40 age groups were significantly lower than those of the other two groups (P37–40 vs P41–47: $p=0.002$; P37–40 vs Adult: $p=0.001$. See *Table 1* for the full test details and all statistical results. All data selection criteria are outlined in the Materials and methods. Uneven sample sizes were considered in the choice of statistical tests and are discussed in Materials and methods as well).

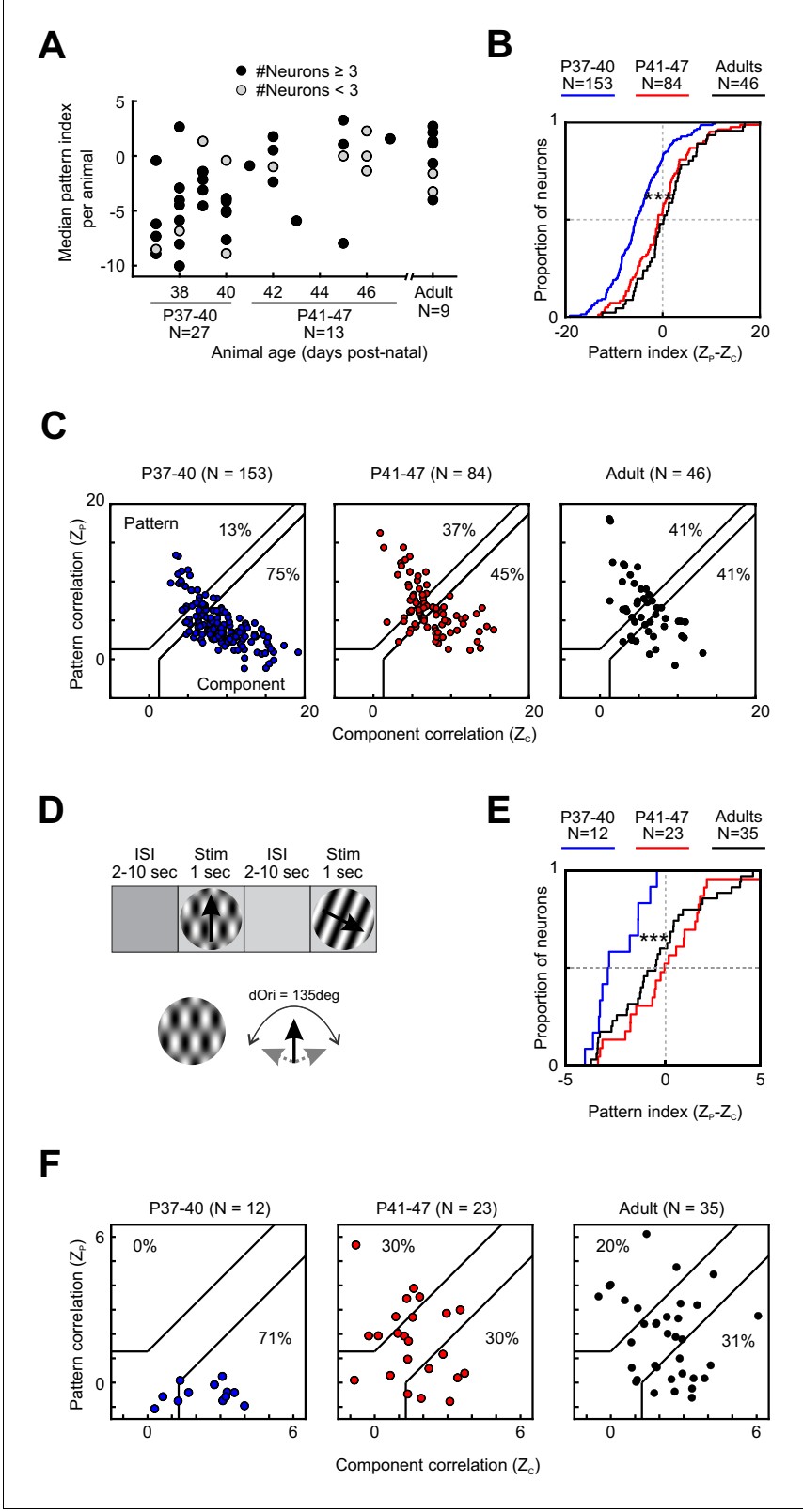

**Figure 2.** Motion integration in PSS matures around P41. (**A**) Median PSS pattern index for each animal included in the study, plotted as a function of age. Gray dots represent animals that yielded one to two neurons, black dots represent animals with data for at least three neurons. (**B**) Cumulative distributions for pattern indices in different age groups. For further information on statistical tests for this and all following plots, see *Table 1*. \*\*\*p<0.001. (**C**)

*Figure 2 continued on next page*

*Figure 2 continued*

Pattern versus component correlations for PSS neurons in animals from different age groups. For each neuron, $Z_P$ is plotted against $Z_C$. Black lines indicate the category boundaries used to classify cells into pattern, unclassified, and component cells. Percentages indicate the portion of neurons falling into the component and pattern categories. (D) Top: Illustration of the classic stimulus presentation paradigm, in which stimuli were presented for 1 s, and stimulus presentations were interleaved by 2–10 s of inter-stimulus interval. Bottom: Example plaid stimulus with 135 degrees of difference between the component directions. (E) Same as (B), but for pattern indices computed from responses to plaids presented in the classic stimulus paradigm as shown in (D). ***p<0.001. (F) Same as (C), but for pattern and component correlations computed based on the classic stimulus paradigm as shown in (D).

The online version of this article includes the following source data and figure supplement(s) for figure 2:

**Source data 1.** Neuron-by-neuron metrics for *Figure 2* and *Figure 2—figure supplement 1*.
**Figure supplement 1.** Changes in pattern and component correlation contribute to the developmental increase in pattern index.

---

Next, we assessed the motion integration capacity of these three groups at the population level by comparing responses across all of the neurons collected for each group (rather than the animal-by-animal analysis used for *Figure 2A*). *Figure 2B* plots the cumulative distribution of pattern indices for each group. Consistent with the trends shown in *Figure 2A*, the pattern index increased significantly between P37–40 and P41–47, at which point it became adult-like (P37–40 vs P41–47: p<0.001. See *Table 1* for the other p values. Non-significant results are omitted from the results text, but are listed in *Table 1*). This change in pattern index was driven by opposite changes in both of the underlying partial correlation coefficients (*Figure 2—figure supplement 1*): Between the two younger age groups, $Z_C$ decreased significantly (p<0.001), while $Z_P$ increased (p<0.001). Changes in pattern and component cell proportions show the same trends (*Figure 2C*): The proportion of component cells decreased with age (75% at P37–40 vs 41% in adults), while the proportion of pattern cells increased (13% at P37–40 vs 41% in adults).

A possible concern regarding these findings is the short presentation time per stimulus in the streaming stimulus paradigm. Fast stimulus presentation times combined with potentially slower motion integration in younger animals might result in pattern integration remaining undetected. In addition, the rapid succession of stimuli in the streaming paradigm might differentially affect PSS across development. As a control, we therefore recorded responses to plaids and gratings presented individually and for longer durations (1 s) at P37–40, at P41–47, and in adulthood. Plaids were restricted to a dOri value of 135 deg in these experiments (see Plaids, classic stimulus presentation in Materials and methods for full stimulus description). The control experiment confirmed our main findings with the streaming stimulus paradigm (*Figure 2D–F*): As before, the pattern index increased significantly between P37–40 and P41–47 (p<0.001), at which point it became adult-like. The number of pattern cells followed the same trend. Thus, the initially lower motion integration levels at P37–40 are not simply a consequence of the streaming stimulus paradigm.

Another possible concern regarding the data is that weaker responses in younger animals may mask motion integration. More precisely, higher variability in responses, particularly at younger ages, might make pattern and component correlation indices more similar, thereby lowering the absolute value of the pattern index. Indeed, for plaids with dOri = 90 deg (which generally evoked strong responses; see section on Relative plaid response strength below) the median response rate was 2.46 Hz at P37–40, but 2.97 Hz in the P41–47 group (rank-sum test, p=0.02). To test whether these response rate differences could contribute to the low pattern indices in the youngest age group, we subsampled the P37–40 data set by selecting only the neurons with firing rates higher than the median at P41–47 (i.e., firing rates larger than 2.97 Hz). For the 56 neurons that fell into this category, the median pattern index remained low at −6.03 (versus −5.6 in the full data set). The significant difference in pattern index between P37–40 and P41–47 also remained for the subsampled data set (Welch's t-test, p<0.001). It therefore seems unlikely that low response rates can explain the lack of motion integration in the youngest age group.

In summary, PSS motion integration – as reflected in the responses to the pattern motion of coherent plaids – developed after direction selectivity and matured about 7 weeks after birth or 1 week after direction selectivity (which also corresponds to about 2 weeks after eye opening).

**Table 1.** Extended information for all statistical analyses.

| Figure | Experimental groups | Metric or variable | Test | p-value | Other stats |
|---|---|---|---|---|---|
| *Figure 1* Sup 1 | P37 vs adult | Direction selectivity index | Welch's t-test | 0.86 | d' = 0.04 |
| | | | | | |
| *Figure 2A* | P37–40 vs P41–47 | Animal median pattern index | Welch's t-test | 0.002 | d' = 1.1 |
| *Figure 2A* | P37–40 vs adult | Animal median pattern index | Welch's t-test | 0.001 | d' = 1.4 |
| *Figure 2A* | P41–47 vs adult | Animal median pattern index | Welch's t-test | 0.59 | d' = 0.23 |
| *Figure 2A* | P37–40 vs P41–47 | Animal median pattern index | Resample test | <0.001 | |
| *Figure 2A* | P37–40 vs adult | Animal median pattern index | Resample test | <0.001 | |
| | | | | | |
| *Figure 2B* | P37–40 vs P41–47 | Pattern index. Multiple dOri | Welch's t-test | <0.001 | d' = 0.74 |
| *Figure 2B* | P37–40 vs adult | Pattern index. Multiple dOri | Welch's t-test | <0.001 | d' = 0.96 |
| *Figure 2B* | P41–47 vs adult | Pattern index. Multiple dOri | Welch's t-test | 0.3 | d' = 0.19 |
| *Figure 2B* | P37–40 vs P41–47 | Pattern index. Multiple dOri | Resample test | <0.001 | |
| *Figure 2B* | P37–40 vs adult | Pattern index. Multiple dOri | Resample test | <0.001 | |
| | | | | | |
| *Figure 2* Sup 1 | P37–40 vs P41–47 | $Z_p$. Multiple dOri | Welch's t-test | <0.001 | d' = 0.74 |
| *Figure 2* Sup 1 | P37–40 vs adult | $Z_p$. Multiple dOri | Welch's t-test | <0.001 | d' = 1.0 |
| *Figure 2* Sup 1 | P41–47 vs adult | $Z_p$. Multiple dOri | Welch's t-test | 0.24 | d' = 0.22 |
| *Figure 2* Sup 1 | P37–40 vs P41–47 | $Z_p$. Multiple dOri | Resample test | <0.001 | |
| *Figure 2* Sup 1 | P37–40 vs adult | $Z_p$. Multiple dOri | Resample test | <0.001 | |
| | | | | | |
| *Figure 2* Sup 1 | P37–40 vs P41–47 | $Z_c$. Multiple dOri | Welch's t-test | <0.001 | d' = 0.61 |
| *Figure 2* Sup 1 | P37–40 vs adult | $Z_c$. Multiple dOri | Welch's t-test | <0.001 | d' = 0.73 |
| *Figure 2* Sup 1 | P41–47 vs adult | $Z_c$. Multiple dOri | Welch's t-test | 0.58 | d' = 0.10 |
| *Figure 2* Sup 1 | P37–40 vs P41–47 | $Z_c$. Multiple dOri | Resample test | <0.001 | |
| *Figure 2* Sup 1 | P37–40 vs adult | $Z_c$. Multiple dOri | Resample test | <0.001 | |
| | | | | | |
| *Figure 2E* | P37–40 vs P41–47 | Pattern index. dOri 135 deg | Welch's t-test | <0.001 | d' = 1.3 |
| *Figure 2E* | P37–40 vs adult | Pattern index. dOri 135 deg | Welch's t-test | <0.001 | d' = 0.94 |
| *Figure 2E* | P41–47 vs adult | Pattern index. dOri 135 deg | Welch's t-test | 0.43 | d' = 0.21 |
| *Figure 2E* | P37–40 vs P41–47 | Pattern index. dOri 135 deg | Resample test | <0.001 | |
| *Figure 2E* | P37–40 vs adult | Pattern index. dOri 135 deg | Resample test | 0.005 | |
| | | | | | |
| *Figure 3A* | V4 vs V5 | Median pattern index | Welch's t-test | 0.005 | d' = 1.8 |
| *Figure 3A* | V4 vs V6 | Median pattern index | Welch's t-test | 0.02 | d' = 1.3 |
| *Figure 3A* | V5 vs V6 | Median pattern index | Welch's t-test | 0.89 | d' = 0.07 |
| *Figure 3C* | V4 vs V5–6 | Pattern index | Welch's t-test | 0.01 | d' = 0.5 |
| *Figure 3C* | V4 vs V5–6 | Pattern index | Resample test | 0.03 | |
| *Figure 3D* | V4 vs V5–6 | $Z_p$ | Welch's t-test | 0.03 | d' = 0.42 |
| *Figure 3D* | V4 vs V5–6 | $Z_p$ | Resample test | 0.04 | |

*Table 1 continued on next page*

*Table 1 continued*

| Figure | Experimental groups | Metric or variable | Test | p-value | Other stats |
|---|---|---|---|---|---|
| *Figure 3D* | V4 vs V5–6 | $Z_c$ | Welch's t-test | 0.02 | d' = 0.52 |
| *Figure 3D* | V4 vs V5–6 | $Z_c$ | Resample test | 0.03 | |
| | | | | | |
| *Figure 4B* | P37–40 vs P41–47 | Relative plaid responses. Wilson–Hilferty transform | ANOVA. Var: age. | 0.80 | F = 0.07 |
| *Figure 4B* | P37–40 vs P41–47 | Relative plaid responses. Wilson–Hilferty transform. | ANOVA. Var: dOri | <0.001 | F = 60 |
| *Figure 4B* | P37–40 vs P41–47 | Relative plaid responses. Wilson–Hilferty transform. | ANOVA. Var: interaction | 0.006 | F = 3 |
| *Figure 4B* | P37–40 vs adult | Relative plaid responses. Wilson–Hilferty transform. | ANOVA. Var: age. | 0.08 | F = 3.1 |
| *Figure 4B* | P37–40 vs adult | Relative plaid responses. Wilson–Hilferty transform. | ANOVA. Var: dOri | <0.001 | F = 44 |
| *Figure 4B* | P37–40 vs adult | Relative plaid responses. Wilson–Hilferty transform. | ANOVA. Var: interaction | <0.001 | F = 4.2 |
| *Figure 4B* | P41–47 vs adult | Relative plaid responses. Wilson–Hilferty transform. | ANOVA. Var: age. | 0.16 | F = 2 |
| *Figure 4B* | P41–47 vs adult | Relative plaid responses. Wilson–Hilferty transform. | ANOVA. Var: dOri | <0.001 | F = 51 |
| *Figure 4B* | P41–47 vs adult | Relative plaid responses. Wilson–Hilferty transform. | ANOVA. Var: interaction | 0.64 | F = 0.7 |
| | | | | | |
| *Figure 4B* | P37–40 vs P41–47 | Relative plaid responses. dOri: 45 deg. Wilson–Hilferty transform. | Welch's t-test | 0.16 | d'=0.19 |
| *Figure 4B* | P37–40 vs adult | Relative plaid responses. dOri: 45 deg. Wilson–Hilferty transform. | Welch's t-test | 0.005 | d' = 0.58 |
| *Figure 4B* | P41–47 vs adult | Relative plaid responses. dOri: 45 deg. Wilson-Hilferty transform. | Welch's t-test | 0.06 | d' = 0.39 |
| *Figure 4B* | P37–40 vs adult | Relative plaid responses. dOri: 45 deg. Wilson–Hilferty transform. | Resample test | <0.001 | |
| | | | | | |
| *Figure 4B* | P37–40 vs P41–47 | Relative plaid responses. dOri: 157 deg. Wilson–Hilferty transform. | Welch's t-test | 0.02 | 0.32 |
| *Figure 4B* | P37–40 vs adult | Relative plaid responses. dOri: 157 deg. Wilson–Hilferty transform. | Welch's t-test | 0.01 | 0.42 |
| *Figure 4B* | P41–47 vs adult | Relative plaid responses. dOri: 157 deg. Wilson–Hilferty transform. | Welch's t-test | 0.53 | 0.12 |
| *Figure 4B* | P37–40 vs P41–47 | Relative plaid responses. dOri: 157 deg. Wilson–Hilferty transform. | Resample test | 0.002 | |
| *Figure 4B* | P37–40 vs adult | Relative plaid responses. dOri: 157 deg. Wilson–Hilferty transform. | Resample test | <0.001 | |
| | | | | | |
| *Figure 4C* | P37–40 | Pattern index vs relative plaid response. Wilson–Hilferty transform | Pearson correlation | 0.009 | r = 0.21 |
| *Figure 4C* | P41–47 | Pattern index vs relative plaid response. Wilson–Hilferty transform. | Pearson correlation | <0.001 | r = 0.46 |
| *Figure 4C* | Adult | Pattern index vs relative plaid response. Wilson–Hilferty transform | Pearson correlation | <0.001 | r = 0.60 |
| *Figure 4C* | P47–40 vs adult | Pattern index vs relative plaid response. Wilson–Hilferty transform | Correlation difference | 0.02 | z = 2.06 |
| *Figure 4C* | P47–40 vs P41–47 | Pattern index vs relative plaid response. Wilson–Hilferty transform | Correlation difference | 0.003 | z = 2.77 |
| | | | | | |
| *Figure 4 Sup 1C* | P37–40 | Pattern index vs relative plaid response. | Pearson correlation | 0.003 | r = 0.24 |
| *Figure 4 Sup 1C* | P41–47 | Pattern index vs relative plaid response. | Pearson correlation | <0.001 | r = 0.46 |
| *Figure 4 Sup 1C* | Adult | Pattern index vs relative plaid response. | Pearson correlation | <0.001 | r = 0.58 |

*Table 1 continued on next page*

*Table 1 continued*

| Figure | Experimental groups | Metric or variable | Test | p-value | Other stats |
|---|---|---|---|---|---|
| *Figure 5B* | P37–40 vs P44–47 | Relative plaid responses. Wilson–Hilferty transform | ANOVA. Var: age. | 0.001 | F = 11 |
| *Figure 5B* | P37–40 vs P44–47 | Relative plaid responses. Wilson–Hilferty transform. | ANOVA. Var: dOri | <0.001 | F = 11 |
| *Figure 5B* | P37–40 vs P44–47 | Relative plaid responses. Wilson–Hilferty transform. | ANOVA. Var: interaction | 0.26 | F = 1.3 |
| *Figure 5B* | P37–40 vs adult | Relative plaid responses. Wilson–Hilferty transform. | ANOVA. Var: age. | <0.001 | F = 82 |
| *Figure 5B* | P37–40 vs adult | Relative plaid responses. Wilson–Hilferty transform. | ANOVA. Var: dOri | <0.001 | F = 6.8 |
| *Figure 5B* | P37–40 vs adult | Relative plaid responses. Wilson-Hilferty transform. | ANOVA. Var: interaction | 0.10 | F = 1.8 |
| *Figure 5B* | P44–47 vs adult | Relative plaid responses. Wilson–Hilferty transform. | ANOVA. Var: age | <0.001 | F = 100 |
| *Figure 5B* | P44–47 vs adult | Relative plaid responses. Wilson–Hilferty transform. | ANOVA. Var: dOri | <0.001 | F = 7 |
| *Figure 5B* | P44–47 vs adult | Relative plaid responses. Wilson–Hilferty transform. | ANOVA. Var: interaction | 0.03 | F = 2.3 |
| | | | | | |
| *Figure 5D* | P37–40 vs P44–47 | Pattern index. dOri 135 deg | Welch's t-test | <0.001 | d' = 0.61 |
| *Figure 5D* | P37–40 vs adult | Pattern index. dOri 135 deg | Welch's t-test | 0.94 | d' = 0.01 |
| *Figure 5D* | P44–47 vs adult | Pattern index. dOri 135 deg | Welch's t-test | <0.001 | d' = 0.64 |
| *Figure 5D* | P44–47 vs adult | Pattern index. dOri 135 deg | Resample test | <0.001 | |
| | | | | | |
| *Figure 6A* | P44–47 vs muscimol | Relative plaid responses. Wilson–Hilferty transform | ANOVA. Var: Muscimol. | <0.001 | F = 17 |
| *Figure 6A* | P44–47 vs muscimol | Relative plaid responses. Wilson–Hilferty transform. | ANOVA. Var: dOri | <0.001 | F = 4.5 |
| *Figure 6A* | P44–47 vs muscimol | Relative plaid responses. Wilson–Hilferty transform. | ANOVA. Var: interaction | 0.44 | F = 0.97 |
| *Figure 6C* | P44-47 vs ACSF | Relative plaid responses. Wilson–Hilferty transform. | ANOVA. Var: ACSF. | 0.56 | F = 0.32 |
| *Figure 6C* | P44–47 vs ACSF | Relative plaid responses. Wilson–Hilferty transform. | ANOVA. Var: dOri | <0.001 | F = 8.3 |
| *Figure 6C* | P44–47 vs ACSF | Relative plaid responses. Wilson–Hilferty transform. | ANOVA. Var: interaction | 0.98 | F = 0.20 |
| *Figure 6A* | P37–40 vs muscimol | Relative plaid responses. Wilson–Hilferty transform | ANOVA. Var: Muscimol. | 0.04 | F = 4.3 |
| *Figure 6A* | P37–40 vs muscimol | Relative plaid responses. Wilson–Hilferty transform. | ANOVA. Var: dOri | 0.007 | F = 3.0 |
| *Figure 6A* | P37–40 vs muscimol | Relative plaid responses. Wilson–Hilferty transform. | ANOVA. Var: interaction | 0.94 | F = 0.29 |
| *Figure 6C* | P37–40 vs ACSF | Relative plaid responses. Wilson–Hilferty transform. | ANOVA. Var: ACSF. | <0.001 | F = 12 |
| *Figure 6C* | P37–40 vs ACSF | Relative plaid responses. Wilson–Hilferty transform. | ANOVA. Var: dOri | <0.001 | F = 6.6 |
| *Figure 6C* | P37–40 vs ACSF | Relative plaid responses. Wilson–Hilferty transform. | ANOVA. Var: interaction | 0.42 | F = 1 |
| | | | | | |
| *Figure 6B* | P44–47 vs muscimol | Pattern index | Welch's t-test | 0.01 | d' = 0.52 |
| *Figure 6D* | P44–47 vs ACSF | Pattern index | Welch's t-test | 0.4 | d' = 0.15 |
| *Figure 6B* | P37–41 vs muscimol | Pattern index | Welch's t-test | 0.71 | d' = 0.07 |
| *Figure 6D* | P37–41 vs ACSF | Pattern index | Welch's t-test | 0.02 | d' = 0.46 |
| *Figure 6B* | P44–47 vs muscimol | Pattern index | Resample test | 0.02 | |
| *Figure 6B* | P37–41 vs ACSF | Pattern index | Resample test | 0.01 | |
| | | | | | |
| *Figure 7C* | P37–40 vs P44–47 | Model-data correlation | Rank-sum test | 0.14 | |
| *Figure 7E* | P37–40 vs P44–47 | Model excitation ($W_{exc}$) | Rank-sum test | 0.06 | |
| *Figure 7E* | P37–40 vs P44–47 | Model inhibition ($W_{inh}$) | Rank-sum test | <0.001 | |
| *Figure 7E* | P37–40 vs P44–47 | Model PSS threshold ($T_{PSS}$) | Rank-sum test | 0.96 | |
| *Figure 7E* | P37–40 vs P44–47 | Model inhibition ($W_{inh}$) | Resample test | <0.001 | |

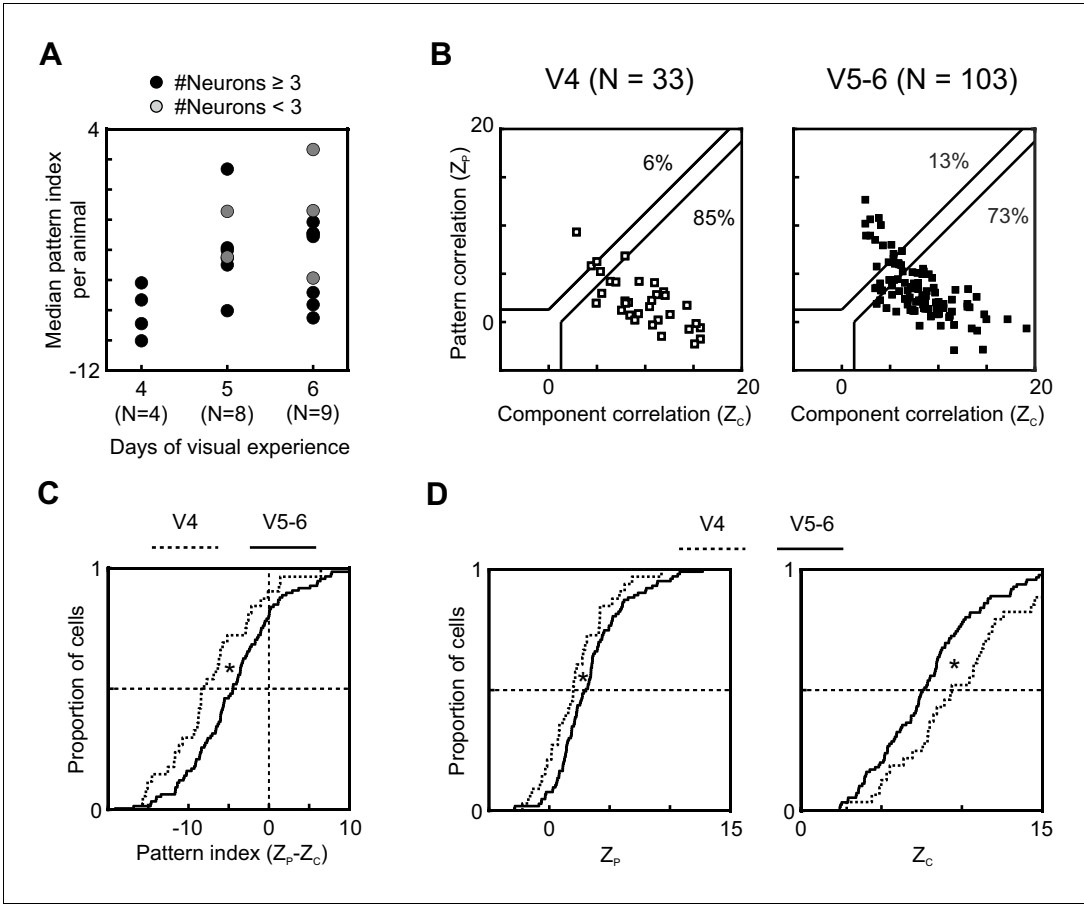

**Figure 3.** Longer visual experience results in a higher degree of PSS motion integration. (**A**) Median PSS pattern index for each animal aged P37–40, plotted as a function of visual experience. Gray dots represent animals that yielded 1–2 neurons, and black dots represent animals with data for at least three neurons. (**B**) Pattern versus component correlations at P37–40 split according to visual experience (same plot format as *Figure 2C*). (Left) Animals with 4 days of visual experience. (Right) Animals with 5–6 days of visual experience. (**C**) Cumulative distributions of pattern indices for PSS cells in kits with 4 days (V4, dashed line) or 5–6 days (V5-6, solid line) of visual experience. *p<0.05. (**D**) Cumulative distributions for component correlation (left) and pattern correlation (right) for the two groups. *p<0.05.

The online version of this article includes the following source data for figure 3:

**Source data 1.** Neuron-by-neuron metrics for *Figure 3*.

## Impact of visual experience on motion integration development

Visual experience has been shown to play a key role in the development of direction selectivity in ferret V1 (*Li et al., 2006*; *Popović et al., 2018*; *Van Hooser et al., 2012*). Here, we made use of inter-animal variability in development to begin to probe whether it might similarly impact the development of higher-level motion functions in ferrets. In the previous section, we compared PSS responses strictly by age. However, the age at eye opening – and with it the starting point of patterned visual experience – varies across animals (for this data set: range P31-36). This made it possible to test whether differences in the duration of visual experience resulted in measurable differences in motion integration in animals of the same age. Systematic changes in motion integration due to the amount of visual experience should be most detectable in the youngest age group (P37–40), which have the most immature levels of PSS motion integration, and consequently the largest capacity for improvement. We therefore restricted the analysis to animals of this age range. No animal in this group had less than 4 days of experience. A single animal had 7 days of visual experience at the time of recording and was excluded from further analysis as an outlier. *Figure 3A* plots the median pattern index for each of the remaining animals in the P37–40 age group as a

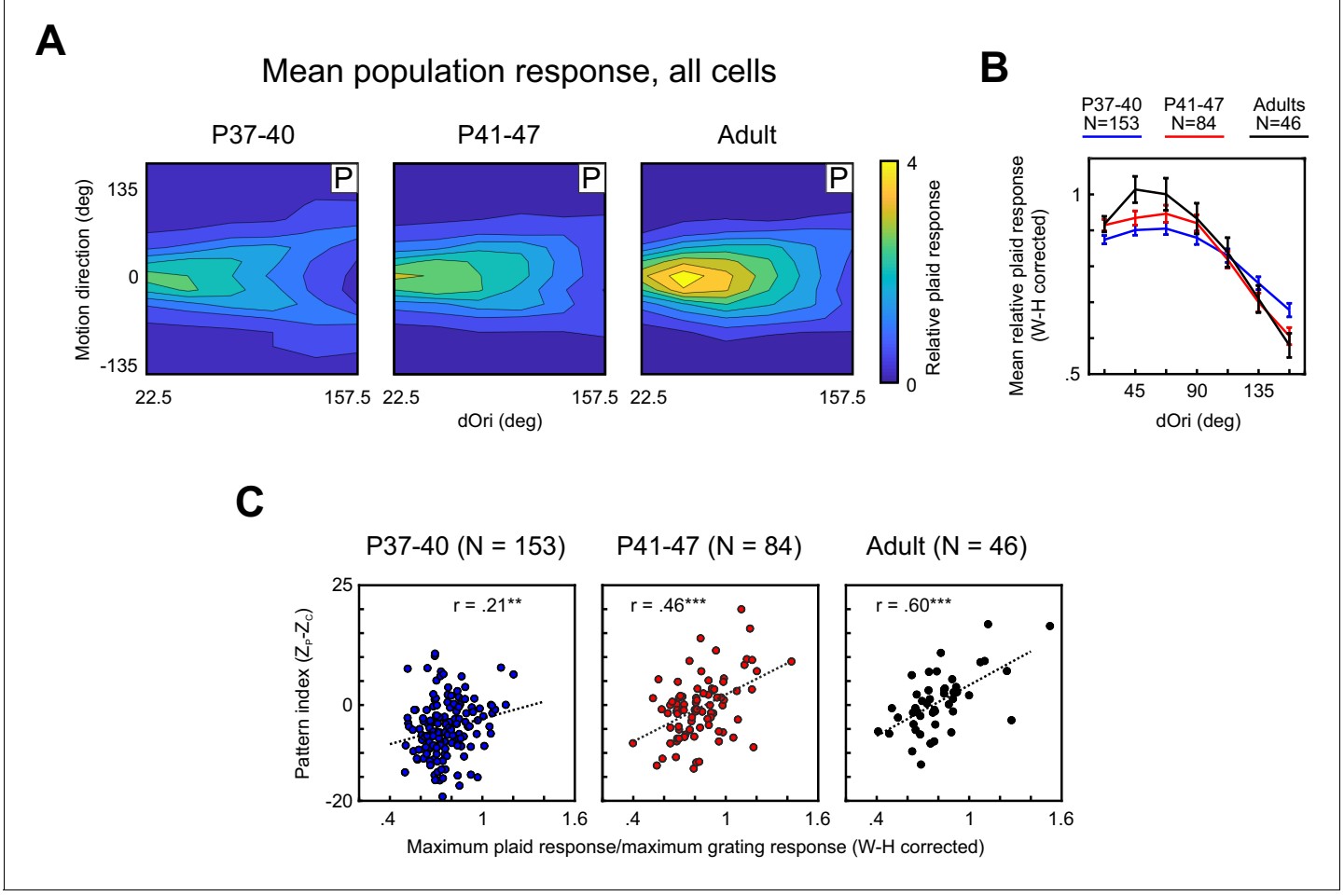

**Figure 4.** Responses of PSS neurons to plaid stimuli become increasingly modulated by dOri with development, especially in pattern cells. (A) Average 2D profile of PSS responses to plaids at different ages (plotted in same format as *Figure 1D–E*). Average response profiles were computed by normalizing the data for each neuron by its mean grating response, shifting responses in direction space to center the preferred direction at 0 deg, and averaging across all neurons in each age group. (B) Average relative plaid response as a function of dOri. The average relative plaid response shown here was computed by averaging the relative plaid responses shown in (A) across all directions at each dOri. All data were transformed using a Wilson–Hilferty transformation to achieve a normal distribution at each dOri value. Error bars: ± SEM. For statistical comparisons of data across age groups, see *Table 1*. (C) Ratio between the maximum response to plaids of dOri = 90 deg and the maximum response to gratings versus the pattern index for all PSS neurons at different ages. Dashed lines indicate linear fits, r the correlation coefficient. Data are shown after a Wilson–Hilferty transformation. **p<0.01. ***p<0.001.

The online version of this article includes the following source data and figure supplement(s) for figure 4:

**Source data 1.** Neuron-by-neuron metrics for *Figure 4* and *Figure 5—figure supplement 1*.

**Figure supplement 1.** Relative plaid responses in PSS and the Wilson–Hilferty transformation.

function of their amount of visual experience at the time of recording (same format as for *Figure 2A*). This figure shows an increase in pattern index with visual experience. In particular, kits with 4 days of visual experience had lower pattern indices than kits with 5 or 6 days of visual experience (V4 vs V5: p=0.004; V4 vs V6: p=0.02). We therefore divided data into two groups, V4 and V5–6, for a population-level analysis (*Figure 3B–D*). Longer visual experience indeed resulted in a larger proportion of pattern cells (V4: 6%, V5–6: 13%) and a significant increase in pattern index (p=0.01). Analyzed separately, both pattern and component correlations showed significant changes ($Z_P$: p=0.03; $Z_C$: p=0.02).

The above analysis relies entirely on natural variation in eye opening across animals. While this offers the advantage of studying the impact of visual experience outside of artificial manipulations, it limits the available data set. For example, certain combinations of gestational age and visual

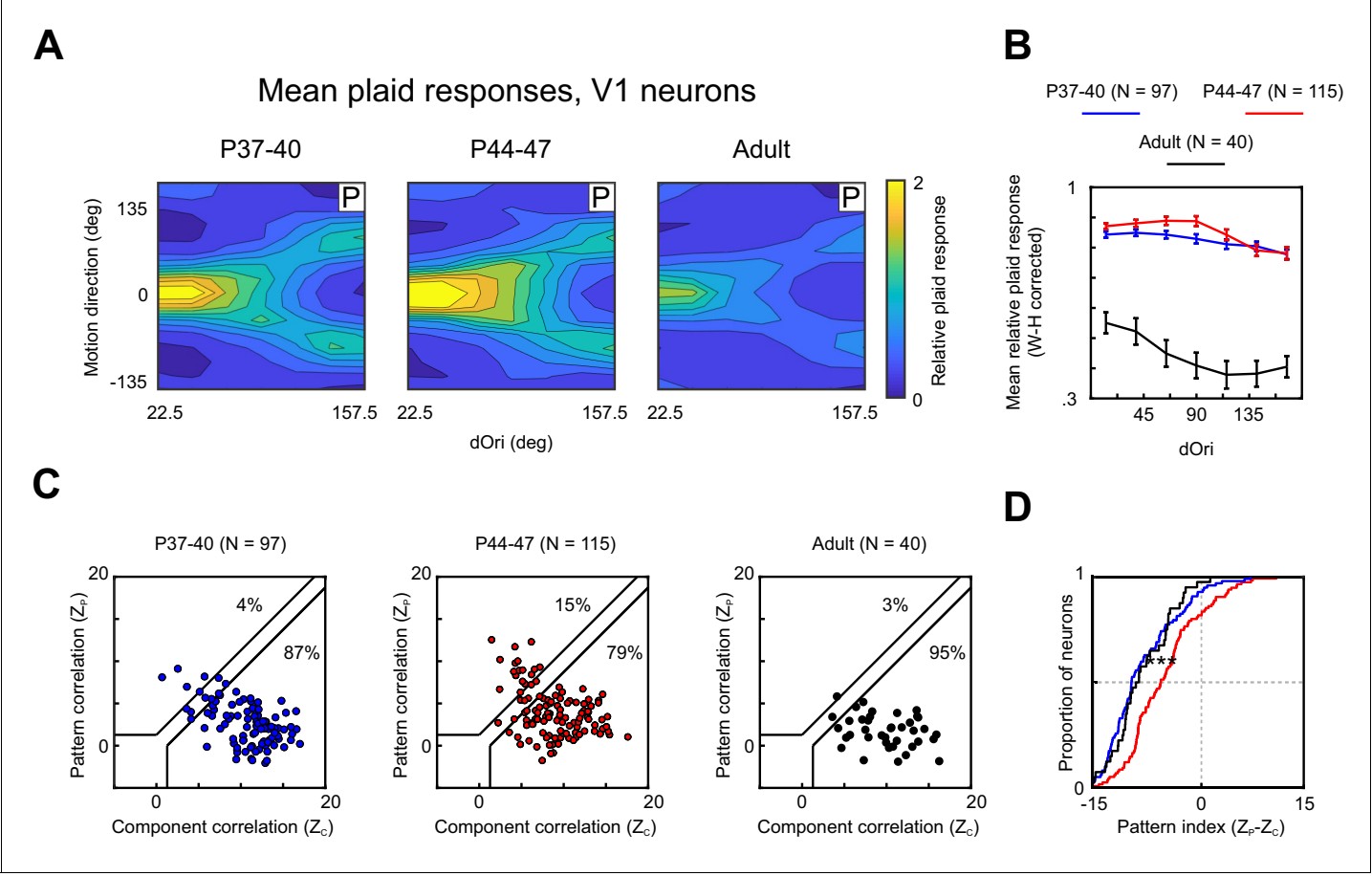

**Figure 5.** V1 responses to plaids change during PSS motion integration development. (A) Average 2D profile of responses to plaids in V1 at different ages (computed and plotted as in *Figure 4A*). (B) Average relative plaid response in V1 as a function of dOri (data shown after Wilson–Hilferty transformation; same format as *Figure 4B*). Error bars: ± SEM. (C) Pattern versus component correlations for V1 neurons in different age groups (same format as *Figure 2C*). (D) Cumulative distribution plots for V1 pattern indices at different ages. Same N and color scheme as (C).
The online version of this article includes the following source data and figure supplement(s) for figure 5:

**Source data 1.** Neuron-by-neuron metrics for *Figure 5* and *Figure 5—figure supplement 1*.
**Figure supplement 1.** Motion integration levels in V1 are lower than those in PSS across development.

experience duration did not occur in our data set. In addition, some correlation between eye opening and gestational date is unavoidable and resulted in the fact that animals in the V4 group were slightly younger (mean age 37.6 days, range 37–40) than animals in the V5–6 group (mean age 38.6, range 37–40). We therefore tested whether visual experience indeed had an effect in addition to age by performing a two-way ANOVA on the pattern index data from all PSS cells with factors visual experience (V4–V6) and age (P37–40). Main effects for both factors were significant (main effect age: p=0.02; main effect visual experience: p=04), demonstrating that both contributed significantly to the observed variance in pattern index across cells. Due to the sampling limitations imposed by natural variability, additional experiments will be needed to fully disentangle the impacts of age and experience. Yet, our results suggest that visual experience impacts the development of PSS motion integration, in addition to the contribution of age.

## Developmental changes in PSS plaid response strength

Our stimulus set allowed us to probe PSS plaid responses not just as a function of plaid direction, but also as a function of dOri, the intersection angle between the two component gratings. Possible reasons for a response modulation with dOri include differences in the spatial frequency of these plaids, as well as differences in their speed (*Zaharia et al., 2019*; *Priebe et al., 2003*). Additionally,

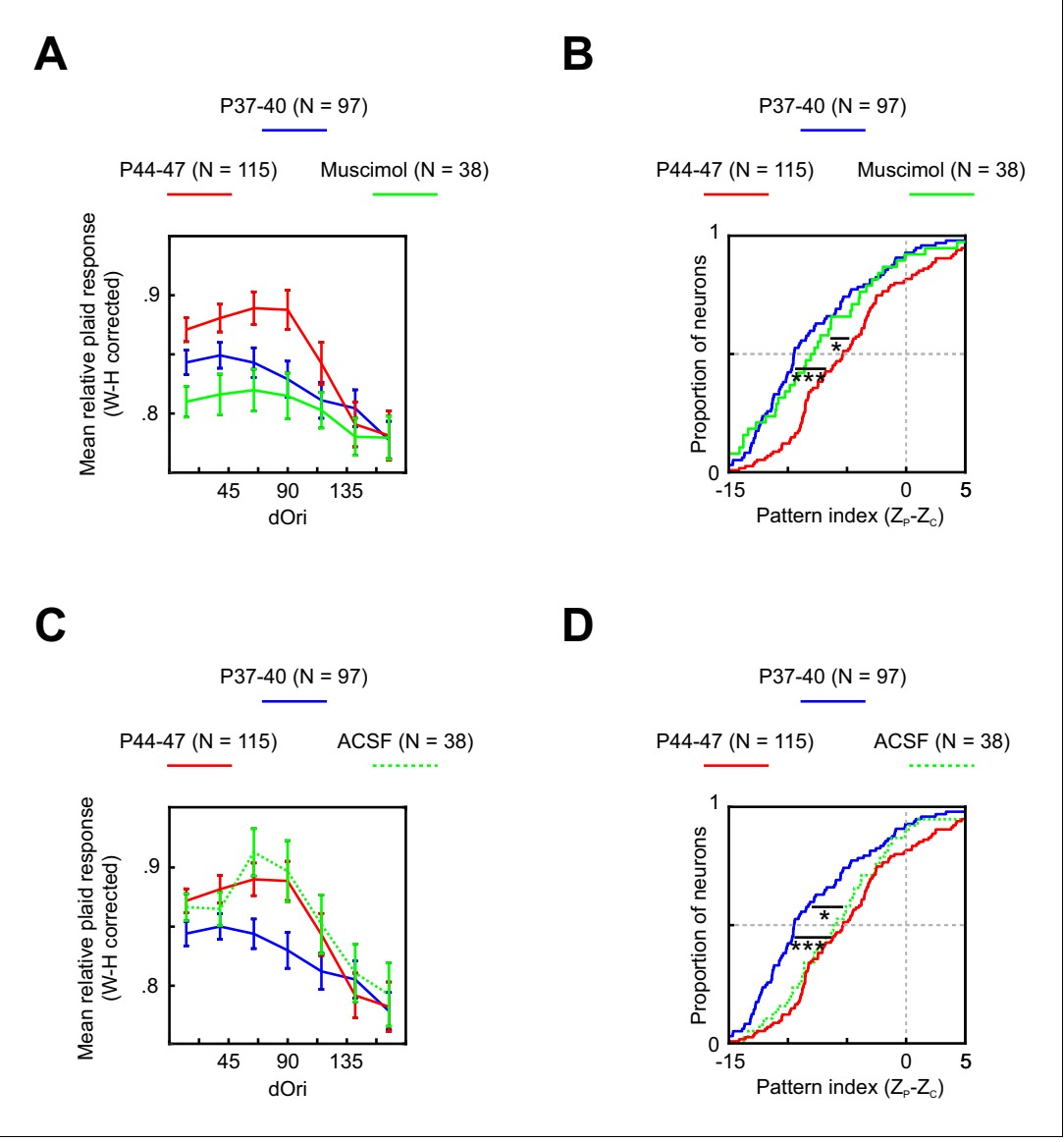

**Figure 6.** Changes in V1 responses at P44–47 depend on feedback from PSS. (A) Average relative plaid response in V1 at P37–40 (same data as *Figure 5B*), P44–47 (same data as *Figure 5B*), and P44–47 with muscimol inactivation of PSS. Error bars indicate ± SEM. Data are shown after a Wilson–Hilferty transformation. For statistical analysis of data across age groups, see *Table 1*. (B) Cumulative distribution of V1 pattern indices at P37–40 (same data as *Figure 5D*), P44–47 (same data as *Figure 5D*), and P44–47 with muscimol inactivation of PSS. For statistical analysis of data across age groups, see *Table 1*. *p<0.05. ***p<0.001. (C, D) Same plots as (A, B) but after a control injection of ACSF into PSS. Note that the ACSF group included one animal aged P48, making the age range for this group P44–48.

The online version of this article includes the following source data for figure 6:

**Source data 1.** Neuron-by-neuron metrics for *Figure 6*.

the motion integration mechanisms acting in PSS will likely have an impact on plaid response strength by determining how strongly different component signals interact, an assumption that is supported by the observation that the strength of plaid responses is positively correlated with pattern index in both primate MT and adult ferret PSS (*Lempel and Nielsen, 2019*; *Wang and Movshon, 2016*).

Here, we found that the dependency of response strength on dOri underwent a series of developmental changes. *Figure 4A* captures these changes by plotting the population tuning profile as a

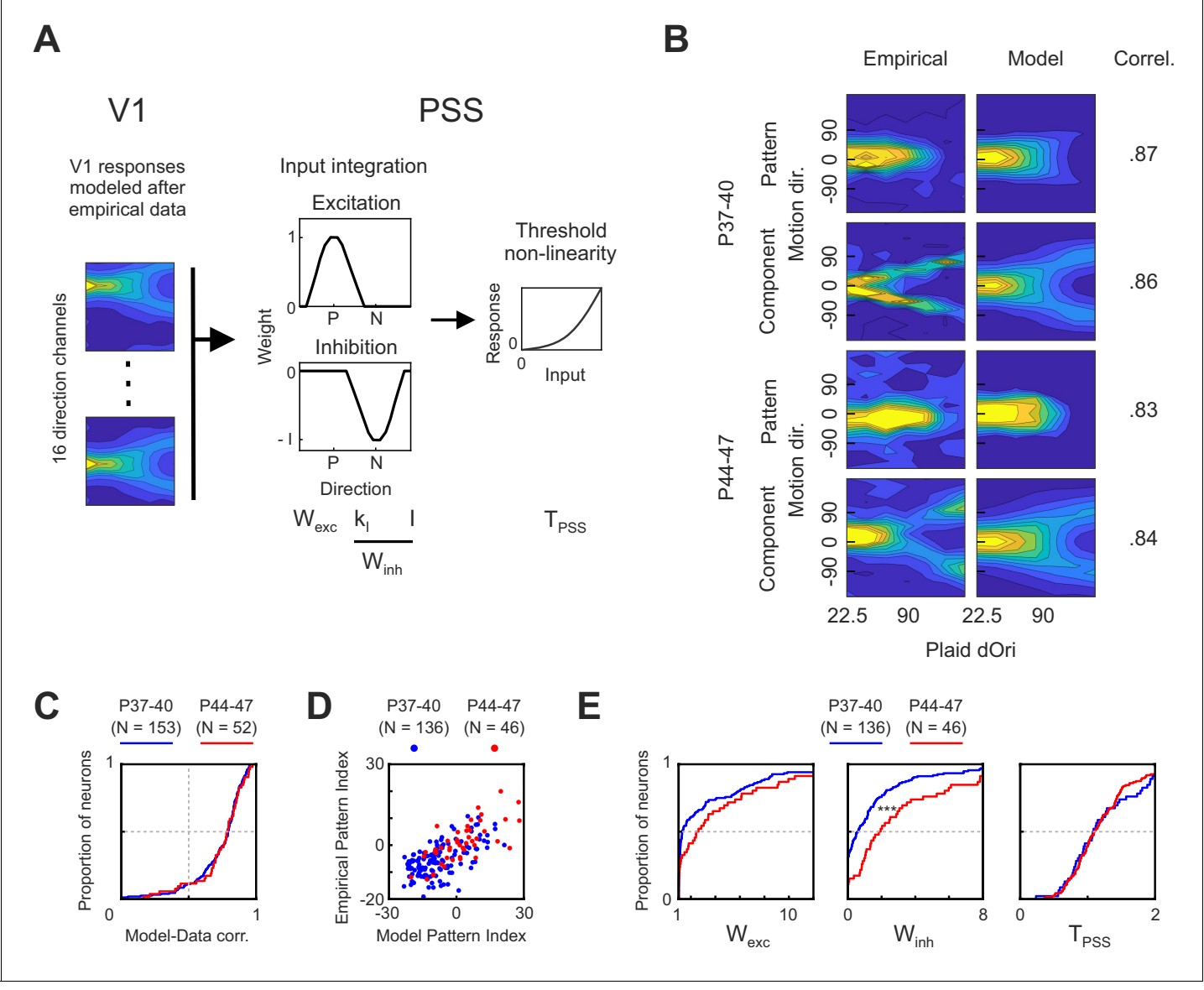

**Figure 7.** A computational model of the ferret motion pathway for testing mechanisms behind motion integration development. (**A**) Diagram of the model used to explain PSS plaid responses across development. The first stage was composed of 16 V1 direction selective cells whose responses were modeled after empirical V1 data from different age groups. Responses from the V1 stage were then integrated in PSS using a combination of an excitatory and an inhibitory weight function (P, preferred direction; N, null direction). Finally, an exponential non-linearity was applied to the PSS responses. The model had four variables, which are listed below the stage to which they belong. For analysis of model results the two variables controlling inhibition ($K_I$ and $I$) where combined into a single metric ($W_{inh}$, see Materials and methods). (**B**) Examples of empirical PSS responses to plaids (left column) and corresponding model PSS responses (right column) after optimizing model parameters to maximize the likelihood of the empirical data for each neuron. Examples show one pattern and one component cell from animals before (P37–40) and after (P44–47) development of motion integration in PSS. Numbers to the right of response profiles indicate correlation between model and empirical responses. (**C**) Distribution of correlation coefficients between model and empirical responses for each PSS neuron in the two age groups. Results from neurons with model-data correlation below 0.5i were excluded from further analyses. (**D**) PSS pattern index for every neuron, computed using empirical data or model data for the neuron, for both age groups. (**E**) Cumulative distribution of all model variables after fitting the model to data from kits before (P37–40) and after (P44–47) the development of motion integration. ***p<0.001.

The online version of this article includes the following source data for figure 7:

**Source data 1.** Neuron-by-neuron metrics for *Figure 7*.

function of dOri and plaid direction for the different age groups, with the plaid responses of every neuron normalized by the response to gratings. We also quantified changes in relative plaid responses by averaging across all plaid directions (*Figure 4B*). Note that when analyzed separately for each dOri, the relative plaid response followed a $\chi^2$ distribution (*Figure 4—figure supplement 1*). We therefore used a Wilson–Hilferty transformation (*Wilson and Hilferty, 1931*) to transform relative plaid responses to a normal distribution before any of the subsequent analyses. *Figure 4B* shows the transformed data; the raw data and effects of transformation are presented for reference in *Figure 4—figure supplement 1*.

Comparing the average relative plaid responses across age groups revealed that their dOri dependency systematically changed, especially between the youngest age group and both older groups. Most notably, in the youngest animals, changes in dOri caused less changes in plaid responses than in older animals. This was reflected in significant interactions between dOri and age when comparing the youngest animals against either of the other two groups using a two-way ANOVA (P37–40 vs P41–47, p=0.006; P37–40 vs Adult, p=0.004). As a post hoc test, we compared relative plaid responses between age groups at the dOri values evoking maximal and minimal responses in adults (dOri = 45 and 157 deg, respectively). At dOri = 45 deg, we observed significantly lower plaid responses in the youngest animals than in adults (p=0.005). Responses were not different from the P41–47 group for this dOri value. In contrast, responses at dOri = 157 deg were significantly higher in the youngest group than either of the other groups (P37–40 vs P41–47, p=0.02; P37–40 vs Adult, p=0.01). Thus, the post hoc tests confirm that the youngest animals showed less modulation of plaid response strength with dOri than older animals. It should be pointed out that the reduction in responses to plaids with large dOri values between P37–40 and P41–47 correlated with a large reduction in component responses (*Figure 4A*). In the youngest age group, the impact of the stronger component responses is visible in the clear 'V'-shape of the population tuning profile. As could be expected, component responses were less affected by dOri, which resulted in a higher response to plaids with large dOri in the youngest age group when averaging across all plaid directions (as done for *Figure 4B*).

As mentioned above, the maximum relative plaid response is positively correlated with the pattern index in ferret PSS and primate MT in adult animals (*Lempel and Nielsen, 2019*; *Wang and Movshon, 2016*). This relationship also emerged with development (*Figure 4C*; note that these data also were transformed using the Wilson-Hilferty transformation): While the correlation coefficient between pattern index and maximum relative plaid response was significant even for the P37–40 group, *Figure 4C* shows the dependency between the two variables was quite weak at this age. The correlation coefficient in the older age groups was significantly higher (P37–40 vs P41–47, p=0.02; P37–40 vs Adult, p=0.003), with a more obvious relationship between the two variables apparent in adults (*Figure 4C*). In summary, not only did the prevalence of pattern responses to plaids increase between P37–40 and P41–47, the dependency of this response on plaid configuration and pattern index developed as well, showing more comprehensive changes in how the visual system handles stimuli consisting of multiple interacting components.

## Developmental changes in V1 plaid responses

Processing of stimuli in V1 is likely an important precursor to PSS functions. V1 does not just compute the local direction signals for the plaid components, the response to these components is modulated by interactions between the plaids through mechanisms like cross-orientation inhibition and motion opponency. These V1 mechanisms could further drive modulations in plaid response strength, in addition to the PSS mechanisms discussed above. We therefore sought to identify developmental changes in V1 plaid processing to gain a more complete picture of the development of the motion pathway as a whole. Accordingly, we recorded V1 responses to plaids at different ages. We selected two age groups for these experiments. The first covered the same age range as the youngest PSS animals, P37–40. The second group covered P44–47, chosen to coincide with the end of the age range of the second PSS group so as to maximize our ability to detect age-dependent differences.

Recording V1 responses to plaids revealed a series of developmental changes. First, we observed changes in the strength of plaid-evoked responses (*Figure 5A,B*). Between the two younger age groups, responses to plaids generally increased independent of dOri (two-way ANOVA, contribution of age p=0.001), before decreasing strongly in adults. With the decrease in adults also came some

changes in the dOri-driven response modulation, as evidenced by significant interactions between dOri and age when comparing the P44–47 age group with the adults using a two-way ANOVA (contribution of age, p<0.001. interaction, p=0.03). The overall decrease in V1 plaid responses was very pronounced in our data set. Note, however, that a previous study (*Popović et al., 2018*) investigating the development of V1 cross-orientation inhibition in ferrets (i.e., looking at responses to plaids with dOri = 90 deg) did not observe changes in plaid responses between P40 and adulthood, something that will need to be resolved with further experiments (see also Discussion).

Second, we made an unexpected discovery in our V1 data set: In adult animals, V1 was strongly dominated by component responses (see *Figure 5E,F* and *Lempel and Nielsen, 2019*). The same held true for the youngest age group: Across the P37–40 animals, we observed a similar number of pattern and component cells as in adults (pattern: 4% versus 3%, component: 87% versus 95%; *Figure 5E*), and the pattern index distributions for both groups closely aligned (*Figure 5F*). However, the middle age group deviated from this pattern. Across animals aged P44–47, we observed a larger number of pattern cells (15%), and the pattern index distribution was significantly shifted toward stronger integration (p<0.001). It should be noted that even with this shift V1 pattern indices remained significantly lower on average than PSS pattern indices (*Figure 5—figure supplement 1*, p<0.001). The same was true also for the two other age groups (p<0.001 for both groups). Taken together, our data suggest a temporary change in V1 plaid responses, characterized by stronger integration of the two component directions and weaker inhibition caused by the simultaneous presence of two gratings. The combination of these two changes had an interesting consequence: V1 responses were enhanced for plaid conditions in which the pattern motion was in the preferred direction, but were left largely unchanged for conditions in which the components moved in the preferred direction (*Figure 5A*). Thus, increases in response strength were largely reserved for pattern responses.

In an effort to identify possible sources for these changes, we next recorded V1 plaid responses at P44–47 while inactivating PSS with the GABA$_A$ agonist muscimol. Inactivating PSS reversed both temporary changes in V1 plaid responses (*Figure 6A,B*). The relative plaid response strength was significantly reduced relative to control data (ANOVA normal versus muscimol p<0.001), reaching the same levels seen normally at P37–41. Similarly, PSS inactivation resulted in a significant shift of the pattern index to lower values (p=0.01) that matched both younger and older animals (see *Table 1* for statistics). A sham PSS injection without muscimol (*Figure 6C,D*) did not impact V1 relative plaid responses or pattern indices (see *Table 1* for statistics), confirming that PSS inactivation, rather than side effects of the injection, caused the observed V1 effects. Thus, both changes in V1 plaid responses are at least partially dependent on feedback from PSS.

## Modeling the relative impact of developmental changes in V1 versus PSS on motion integration

As described in the previous section, we observed temporary changes in V1 plaid responses between P44 and P47. The results of the inactivation experiments indicate that these changes depend on feedback from PSS. At the same time, changes in V1 presumably propagate to PSS (and more generally areas downstream from V1), potentially initiating changes in plaid processing throughout the motion pathway. What consequences could the V1 changes have on motion integration computations in PSS, in particular relative to developmental processes occurring within PSS? Obviously, this question will need to be answered experimentally, but as a first step toward providing some answers we made use of a computational model for the motion pathway.

We have previously shown that a two-stage model consisting of V1-like local direction filters followed by a PSS-like integration stage could fit well to neural responses in PSS (*Lempel and Nielsen, 2019*). Here, we used a modified version of this model to investigate development. Since the goal was to identify the contribution of particular V1 and PSS changes, we replaced the front end of the model. Rather than adjusting the parameters of the V1 stage as needed for the best fit to a given PSS neuron, we fixed it to match the experimentally observed data. More precisely, in the model implemented here (see *Figure 7A*), the first stage was set to be a bank of 16 direction filters that when exposed to plaids would reproduce our measured average V1 responses as closely as possible, including their dOri dependence. Two modifications were made to the empirical V1 response profile in building the model V1 filters: First, we made responses symmetric around the preferred direction. Second, motion pathway models usually assume perfect direction tuning in the V1 stage (i.e., no

responses to stimuli moving in the null direction). We similarly modified the V1 response profile to achieve maximal direction tuning (see Materials and methods for details).

The PSS integration stage of the model remained identical to our previous study. In this stage, the responses of the 16 V1 direction channels were integrated using a weight function with an excitatory component centered at the preferred direction and an inhibitory component centered at the null direction. Both components were modeled as von Mises functions. The width of the excitatory component, as well as the width and amplitude of the inhibitory component were free model parameters. Below, they are summarized as $W_{exc}$ and $W_{inh}$, which refer to the area under the excitatory and inhibitory component, respectively. The integration stage finally was followed by an exponential non-linearity determined by the free parameter $T_{PSS}$. Note that one major purpose of the model was to test whether the observed V1 changes could propagate to PSS (and how they would affect PSS responses). For this reason, the model remained strictly feedforward, despite the above observations that V1 changes depended on feedback from PSS. Future models will need to expand to more complicated models including both feedforward and feedback links to fully explore the dependencies between areas.

As a first modeling exercise, we attempted to identify which PSS-internal mechanisms might explain the developmental changes occurring after P40. To this end, we set the V1 stage to the data measured between either P37–40 or P44–47 and then used a maximum likelihood approach to fit the model to PSS responses measured in the same age range (see Materials and methods). Note that because the V1 data only covered P44–47, we restricted the PSS data set to the same range for the modeling analysis. As illustrated by the example pattern and component cells shown in *Figure 7B*, the model was well able to fit PSS neurons of different types at both ages. This was further confirmed by a generally high correlation between modeled and measured data for both groups, with little differences between the groups (*Figure 7C*). In all subsequent analyses, we excluded cells with correlation coefficients below 0.5. For both age groups predicted and measured pattern indices generally matched well (*Figure 7D*). While the model generally overestimated the absolute value of a neuron's pattern index, likely because of lower noise levels in the modeled responses, pattern cells remained classified as pattern cells based on the model fits, and component cells remained classified as component cells.

The good model fit for both age groups allowed us to compare model parameters across ages. Cumulative distributions for $W_{exc}$, $W_{inh}$ and $T_{PSS}$ are plotted for both age groups in *Figure 7E*. These plots show a trend toward increasing $W_{exc}$ with age, but the difference between age groups did not reach significance (p=0.06). At the same time, $W_{inh}$ significantly increased between the two groups (p<0.001). $T_{PSS}$, finally, remained unchanged. Thus, the model results suggest that changes in PSS null direction inhibition are a major component of motion integration development.

Having identified the major changes internal to PSS that could be driving motion integration development, we next sought to determine how the temporary V1 changes complemented them. To this end, we used the following stepwise procedure: We first initiated the model with its state at P37–40. In this baseline model, the V1 stage was set to match the empirical data for P37–40, as shown in the first column of *Figure 8A*. We also set the parameters for the PSS stage to those determined by fitting the model to PSS data from the same age group. One set of parameters was computed for each neuron in the data set (N = 136), allowing us to construct a population of model PSS neurons. The resulting model PSS responses are summarized in *Figure 8A* by plotting the average plaid response profile for all PSS model neurons (third column), as well as the resulting pattern index distribution (fifth column).

We then used modifications of the baseline model to assess the impact of different developmental changes. First, we asked what contributions the increase in PSS null direction inhibition made to pattern responses. In this model iteration, we kept the V1 stage unchanged relative to the baseline model, that is, it remained set to mimic responses at P37–40. At the same time, we adjusted $W_{inh}$ of the PSS stage to match the levels seen at P40–44. To achieve this, we increased the value of the model parameter I, which controls the strength of inhibition (see Materials and methods), so that the new mean $W_{inh}$ value equaled that determined when fitting the model to data for P40–44 (*Figure 8—figure supplement 1*; note that we did not adjust $K_I$ since it showed little differences between ages in the original fit). *Figure 8B* shows the resulting average PSS plaid response profile (third column) and pattern index distribution relative to that of the baseline model (fifth column). It also shows the response difference between the new model iteration and the baseline version as a function of

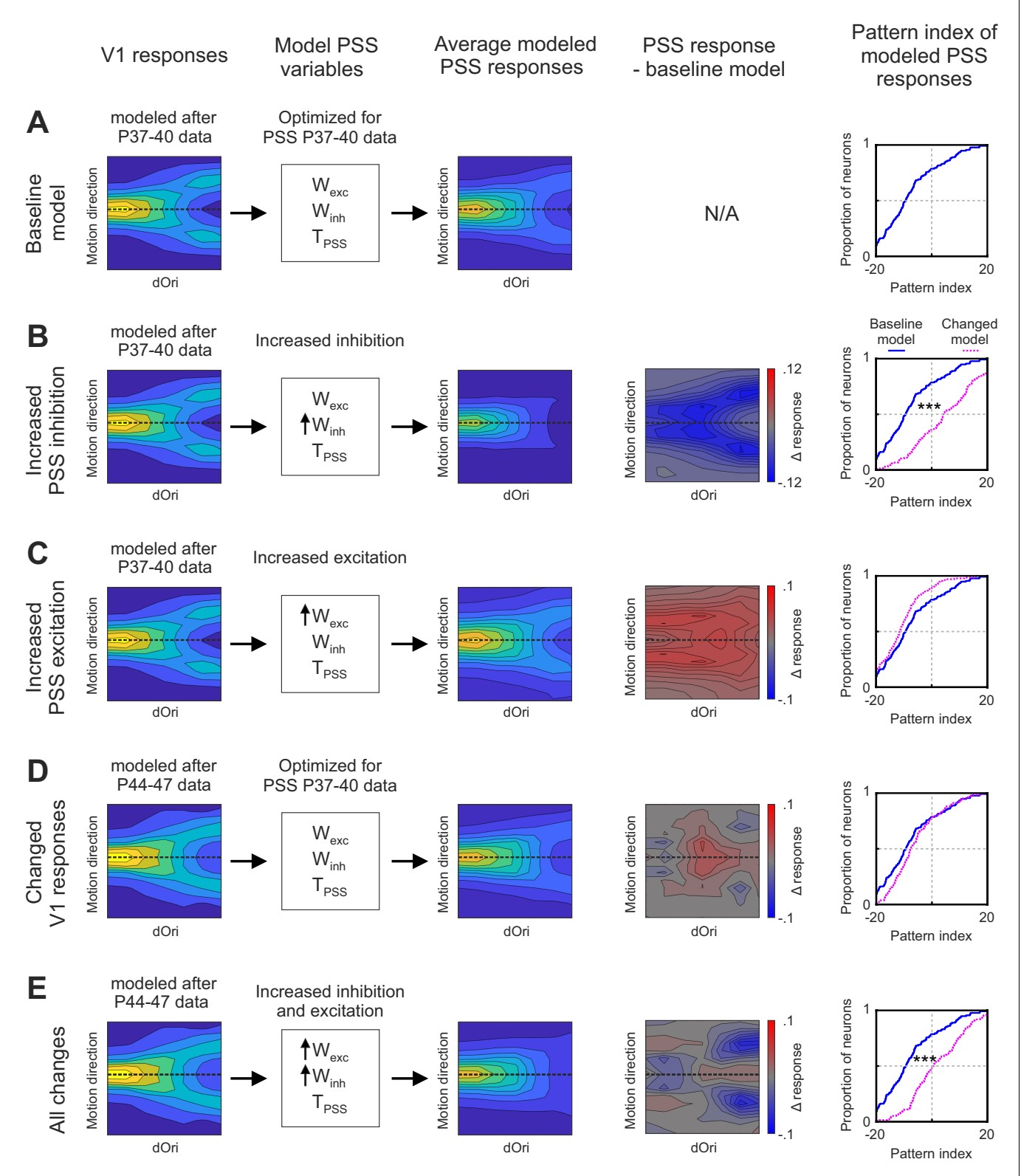

**Figure 8.** Effects of changes in V1 and PSS mechanisms on PSS plaid responses. Summary of different instantiations of the motion pathway model. (A) Baseline model. (B) Model with increased PSS inhibition. (C) Model with increased PSS excitation . (D) Model with V1 responses set to reflect those measured at P44–47. (E) Model combining all changes described for (B), (C), and (D). For each model instantiation, the columns (from left to right) contain the following information: Colum 1: V1 plaid responses used for the model's first stage. Dashed line indicates the preferred motion direction.

*Figure 8 continued on next page*

*Figure 8 continued*

Column 2: State of PSS parameters for motion integration ($W_{exc}$), inhibition ($W_{inh}$), and threshold non-linearity ($T_{PSS}$) relative to the baseline model (determined by fitting the model to data from P37–40). No arrow indicates values were unchanged relative to the baseline model, upwards arrow indicate an increase in the parameter. Column 3: Average plaid responses of model PSS neurons. Dashed line indicates the preferred motion direction. Column 4: Difference between average plaid responses of model PSS neurons in a model instantiation and the average response in the baseline model. Column 5: Cumulative distribution of pattern indices of model PSS neurons for the new model instantiation (dashed pink line) and baseline model (solid blue line). ***p<0.001.

The online version of this article includes the following source data and figure supplement(s) for figure 8:

**Source data 1.** Neuron-by-neuron metrics for *Figure 8* and *Figure 8—figure supplement 1*.
**Figure supplement 1.** Changes in PSS inhibition parameters with development and model parameter settings for the different model instantiations.

plaid direction and dOri (fourth column). These figures demonstrate that while increasing PSS inhibition resulted in a large overall increase in pattern indices, it came with reduced plaid responses. On the one hand, responses were weakened for plaids in which one of the conditions (but not the integrated plaid) moved in the preferred direction (these make up the arms of the 'V'-shaped response profile). Changes to these conditions will increase pattern responses by decreasing component responses and are the key mechanism by which inhibition can shape pattern responses. On the other hand, the increased inhibition extended to conditions in which the pattern movement was in the preferred direction, in particular for intermediate dOri values (likely because for these conditions the components were far enough from the preferred direction to experience some of the increased null direction inhibition). Thus, the improvement in pattern index by increased inhibition came at a cost.

In the second model iteration, we assessed the impact of increasing the excitatory integration of component signals in PSS. While the changes in $W_{exc}$ between P37–40 and P44–47 failed to reach significance, excitation levels nonetheless showed a moderate increase between these two groups. Here, we kept all model parameters with the exception of $K_E$, the only parameter controlling excitation, fixed in their baseline state. $K_E$ was decreased (which has the effect of broadening the excitatory part of the integration function, thereby increasing $W_{exc}$) until the mean of the model $W_{exc}$ distribution matched that determined previously in the P44–47 model fit (*Figure 8—figure supplement 1*). *Figure 8C* again plots the resulting model PSS response to plaids, the difference to the baseline model, and the new pattern index distribution. Increasing excitation actually decreased the pattern index by diminishing differences between pattern and component responses, but as expected, the strength of pattern responses was increased across all dOri values.

In the third model iteration, we then asked what contribution the observed temporary changes in V1 would make to PSS pattern responses. In this iteration, we changed the V1 stage to match the data measured in P44–47 animals, while fixing the PSS model parameters to the values determined for P37–40. The new V1 response profile is plotted in *Figure 8D*, as is the resulting average PSS response profile, its difference relative to the baseline model, and the resulting pattern index distribution. We found that changing the V1 stage on its own had little consequences on PSS pattern indices, which remained largely unchanged from the P37–40 baseline model. However, there were changes in the PSS response to plaids. The selective increases in V1 pattern responses indeed propagated to PSS and resulted in increased responses for pattern motion for plaids with intermediate dOri values.

Finally, we combined V1 and PSS changes in the fourth model iteration to determine their combined effect (*Figure 8E*). The major difference between this model iteration, and a full fit of the pathway model to data from P44 to P47 as performed in *Figure 7*, was the state of the PSS stage. Rather than optimizing these parameters through fitting the model, we only shifted the mean inhibition and excitation levels of the baseline model as described above. In this last model iteration, the increases in inhibition again served to decrease component responses and increase the pattern index. At the same time, increases in PSS excitation, combined with the changes in V1 plaid responses, mitigated the side effects of the increased suppression levels by limiting them to the component responses only.

## Discussion

A hallmark of cortical organization is the existence of specialized pathways dedicated to certain kinds of information processing. Here, we use motion integration – in the form of local and global motion signals to coherent plaids – to characterize the development of the visual motion pathway. On a basic level, our data reveal that motion tuning functions develop according to their complexity: Direction selectivity in V1 and PSS matured before motion integration. This matches findings in the primate, in which MT pattern cells also develop slowly, over the course of months, after direction selectivity develops (*Chino et al., 1997*; *Kiorpes and Movshon, 2014*).

Intriguingly, our data suggest that visual experience has an impact on the development of PSS motion integration. These findings add to an existing body of work on the role of visual experience in the development of the carnivore motion pathway. In both ferrets and cats, visual experience is required for normal development of direction selectivity in V1 (*Li et al., 2008*; *Blakemore and Van Sluyters, 1975*; *Cynader et al., 1976*). At least the basic functions in higher areas are also susceptible to a lack of normal visual experience: Rearing cats in a stroboscopic environment, in which normal motion cues are removed, results in the loss of direction selectivity in higher area PMLS (*Spear et al., 1985*). Our data point to a role of visual experience in the development of the more complex functions of these higher areas as well. Since the animals here were raised under normal conditions, the current data can only show that visual experience can accelerate the development of these functions. Additional experiments are needed to test whether it is actually necessary. However, our findings are consistent with the observation that dark reared cats have marked deficits in a behavioral motion integration task (*Mitchell et al., 2009*). The impact of visual experience on the development of motion integration has also been investigated in the primate. At this time, evidence for a role of visual experience on these functions is mixed: Neurons in MT of amblyopic monkeys show no changes in pattern responses for plaids, but changes in global motion processing were observed when using random dot kinematograms (RDK) (*El-Shamayleh et al., 2010*). It remains to be determined whether a more severe loss of visual experience, as in the carnivore studies, would result in larger deficits in MT functions. Indeed, bilateral congenital cataracts cause large deficits in motion integration performance for RDK in humans, but deficits after monocular cataracts are much smaller (*Ellemberg et al., 2002*).

With the development of motion integration came a change in how the plaid configuration modulated PSS responses. In part, these changes might reflect the development of spatial and temporal tuning functions in PSS, which remain to be studied longitudinally. More generally, it is currently unknown whether PSS neurons are tuned for temporal frequency or speed and how these tuning properties interact with the pattern index, as has been investigated for MT (*Zaharia et al., 2019*; *Priebe et al., 2003*). However, the developmental change in dOri dependency of PSS responses likely reflects at least in part the maturation of motion integration mechanisms. An increase in response to plaids with smaller to intermediate dOri values, for example, is consistent with increasingly stronger interactions between different motion signals that are pooled together to compute pattern motion. This conclusion is further supported by the observation that plaid responses became stronger especially in pattern cells throughout development.

While motion integration mechanisms in PSS are obviously crucial to shaping plaid responses, PSS does not exist in isolation, but receives input from other areas. Our V1 data highlight that processing even in supposedly earlier areas of a pathway should not be treated as mature or static when studying development of a higher area. Here, we found temporary changes in the degree of V1 motion integration around P44–47 and more continuous changes in plaid response strength throughout development. Our model (see also below) shows that these changes might serve an important role in conjunction with developmental changes in PSS. At the minimum, characterizing V1 responses provides a benchmark against which to assess PSS development. Note that our data show a large decrease in V1 plaid responses – that is a large increase in cross-orientation inhibition – between P44–47 and adults. Cross-orientation inhibition has previously been studied for 90 deg plaids in ferret V1, both at P40 and in adult animals (*Popović et al., 2018*). In this previous study, relative plaid responses were reported as the ratio between the preferred plaid and the sum of responses to the components. Using the same metric, we observe similar levels of cross-orientation inhibition around P40 (in the new metric, our data has a mean of 0.21 for plaids with dOri = 90 deg). However, the two data sets diverge strongly for adult animals. While our data show a strong

increase, Popovic et al. report a small but significant decrease in cross-orientation inhibition. This discrepancy may be due to differences in the sampled neuronal population (in terms of laminar location or location within V1), but it is more likely caused by differences in the stimulus presentation protocol (slower presentation of individual stimuli versus fast presentation embedded in a train of stimuli). Resolving these differences will require additional experiments in adult animals, which will also need to address why these differences only occur for adults.

Our model suggests a possible role for the temporary V1 changes we observed. More precisely, the model results show that the changes in V1 responses can complement changes in PSS-internal motion integration mechanisms in a meaningful manner. We modeled PSS motion integration by using a weight function with an excitatory component centered at the preferred direction, and an inhibitory component centered at the null direction. This was motivated by the importance of null direction inhibition in motion computations in ferret V1 (*Wilson et al., 2018*), but it also matches the shape of MT motion integration functions as estimated by fitting a primate motion pathway model to plaid responses (*Rust et al., 2006*). Inhibition proved to be the strongest contributor to improving motion integration. Increasing inhibitory levels from those at P37–40 to the levels at P44–47 strongly increased the pattern index across the entire model neuron population. Our model therefore makes the testable prediction that inhibitory circuits in PSS should mature relatively slowly, not before P40. No data exist for inhibitory circuit development in PSS, but inhibitory circuits in ferret V1 have been found to develop up to P60 both in terms of the relative proportion of different types of inhibitory neurons, as well as their laminar distribution (*Chen et al., 2005*; *Dalva, 2010*; *Gao et al., 1999*; *Gao et al., 2000*). A similarly slow development of inhibitory circuity in PSS does therefore appear possible.

The change in pattern index due to increased inhibition was largely driven by a reduction of component responses. However, the increased inhibitory levels necessarily came at a cost, causing reduction in pattern responses as well. It could be expected that increasing the excitatory integration component in PSS could mitigate these effects. However, there is a limit to how much this component can be changed: The unwanted suppression of pattern responses occurs in particular for plaids with intermediate dOri values. To increase the responses to these plaids, the excitatory component would need to increase in width – but the broader this part of the weight function, the broader the resulting direction tuning function even for simple stimuli like gratings. By dividing the 'workload' of increasing pattern responses between V1 and PSS, the increases in the excitatory weight function in PSS can be kept to a reasonable level. In this context, it is worth reiterating an observation regarding the V1 changes: Because V1 responses became more biased toward pattern responses as they became stronger at P44–47, they boosted pattern responses to the preferred direction only, rather than affecting all conditions. Thus, independent of the exact implementation of the motion pathway model, V1 selectively contributes to strengthening responses to the plaid conditions most important for pattern motion computations.

Additional experiments are required to investigate whether V1 changes are necessary for normal PSS development, how they interact with visual experience, and how other areas beyond V1 and PSS contribute to motion pathway development. The temporary nature of the V1 changes (at least in our data set) also poses interesting questions. If V1 changes are indeed temporary, it would suggest that other mechanisms – originating either in PSS or elsewhere – eventually replace the V1 contribution. Alternatively, V1 changes may actually be permanent, but become limited to the subgroup of neurons that project from V1 to PSS, similar to the other specializations that have been reported for the V1 neurons participating in the motion pathway (*Jarosiewicz et al., 2012*; *Movshon and Newsome, 1996*).

Finally, and most importantly, we have discussed the impact of V1 changes in a purely feedforward context so far. Yet, our inactivation experiments demonstrated that these changes depended on feedback from PSS. Independent of their potential consequences on PSS processing, these findings raise two interesting possibilities (which are not mutually exclusive): In one scenario, PSS actively adjusts V1 signals to (temporarily) aid its own development. In the other, V1 changes are a (more passive) consequence of the change in PSS feedback signals that arises from the newly developed PSS functions. Either scenarios link development across multiple network nodes of the motion pathway, and are important concepts for studying the development of a cortical pathway.

# Materials and methods

## Animal preparation and surgery

All procedures adhered to the guidelines of the National Institute of Health and were approved by the Animal Care and Use Committee at Johns Hopkins University. Experiments were performed in both male and female sable ferrets (*Mustela putoris furo*) aged 37–448 days. Experiments were conducted in anesthetized ferrets, using the same procedures as described in our previous work (*Lempel and Nielsen, 2019*). Briefly, animals were anesthetized during the experiments using isoflurane (during surgery: 1.5–3%, during recording: 0.5–2%) and paralyzed using pancuronium bromide (0.15 mg/kg/hr). A number of vital parameters (heart rate, SpO2, EKG, EtCO2, and the EEG) were monitored continuously to ensure adequate anesthetic depth during the experiments. Neosynephrine and atropine were applied to the eyes to retract the nictitating membrane and dilate the pupil, and animals were fitted with contact lenses. Before recordings, craniotomies were made above either V1 or the posterior bank of the suprasylvian sulcus to reach PSS. We targeted central visual field regions in V1, and central and more peripheral visual field regions in PSS.

## Electrophysiology

Neural signals were recorded using either custom-made tetrodes (12 µm nichrome wire, gold-plated to reach final impedances of 150–500 kΩ) or 64-channel silicon microprobes obtained from the Masmanides lab at UCLA (*Du et al., 2011*). Probes were gold-plated to reach final impedances of 150–300 kΩ. Signals were amplified and recorded using a CerePlex Direct amplifier (Blackrock Microsystems) or a RHD2000 amplifier (Intan Technologies). Raw data were acquired at either 30 or 20 kHz and filtered between 250 Hz and 5 kHz. Spike detection threshold was set manually for each recording based on noise levels. Single-unit isolation was performed off-line using MATLAB (MathWorks). Isolation was based on multiple waveform characteristics (e.g., spike amplitude peak, area under the waveform, repolarization phase slope) recorded on the four tetrode channels or on neighboring channels of the silicon probe. Quality of isolation was confirmed by inter-spike interval (ISI) analysis. Units that displayed ISIs below 1.2 ms were not included in further analyses.

## Muscimol injections

PSS was inactivated by injecting muscimol (2.5 mg/ml in ACSF; Sigma) in multiple sites spanning the posterior bank of the suprasylvian sulcus. For control experiments, ACSF without muscimol was injected instead. To aid the visualization of injection location and spread, 0.05% Fast Green FCF (Fisher Scientific) was added to both the muscimol and control solution. To inject the solution, we used glass pipettes pulled using a micropipette puller (Sutter instrument, model P-1000) and 5 µl calibrated glass capillary tubes (VWR). Infusion was performed at a rate of 0.1–0.5 µl/min. For each animal, injections were performed at four different locations along the posterior bank of the suprasylvian sulcus. At each location, 0.2–0.4 µl of either muscimol or ACSF solution were infused at 4 depths spaced 200–300 µm apart. In total, 4–5 µl of either muscimol of ACSF solution were injected per animal. Recordings were made in V1 1–5 hr after injection. For muscimol experiments, lack of PSS responsiveness was confirmed by extracellular recordings either during or right after the last V1 recording.

## Visual stimuli and experiment design

Visual stimuli were generated using the Psychophysics Toolbox extensions for MATLAB (*Brainard, 1997*; *Pelli, 1997*) and displayed on a 24-inch LCD monitor with a refresh rate of 120 Hz, placed 25–35 cm in front of the ferret. For a subset of experiments, a 43-inch LCD monitor with a refresh rate of 60 Hz was used instead. Monitors were gamma corrected using a SpectraScan 655 (PhotoResearch).

### Measurements of direction selectivity

The direction selectivity data shown in *Figure 1—figure supplement 1* was collected in a set of separate experiments. These experiments consisted of gratings moving in different directions as well as a blank condition. Each condition was repeated five times, with conditions presented in a pseudorandom sequence. Measurements of direction selectivity at P37 used square-wave gratings. In

adults, either square-wave or sine-wave gratings were used to measure direction selectivity. Square-wave gratings were only used for the direction selectivity measurements shown in *Figure 1—figure supplement 1*, and not for any of the subsequent plaid experiments. Grating spatial frequency was set to maximize responses at the recording location (range 0.05–0.1 cycles/deg), as was temporal frequency (range 2–6 Hz). Gratings were shown at 100% contrast and moved in 12 or 16 different directions. Stimuli were presented interspersed with presentation of a gray screen of equal mean luminance. Stimuli were presented for 1 s with inter-stimulus intervals of 2–10 s and at a size of 65 × 50 deg.

### Plaids, classic stimulus presentation

These experiments consisted of plaids and gratings. Gratings in these experiments were sine-wave gratings of optimal spatial and temporal frequency shown at 50% contrast. Plaids were generated by superimposing two 50% contrast sine-wave gratings of the same spatial and temporal frequency at an intersection angle of 135 deg. Both gratings and plaids could move in 16 different directions. Five repetitions of each stimulus condition were shown. Stimulus sizes were optimized for each neuron. For most experiments, stimuli were shown in a circular aperture with radius 8–30 deg, but in a small subset of adult plaid experiments, we used rectangular stimuli of 65 × 50 deg instead. Otherwise stimulus parameters were set as described above.

### Plaids, streaming stimulus presentation

Each trial consisted of a 60 s long sequence of short stimulus presentations (3–4 stimuli/s). Each sequence was preceded and followed by a 2–4 s long presentation of a gray screen of equal luminance. Stimulus sequence was determined randomly by picking from all stimulus conditions with replacement. The following stimulus conditions were used: blank (uniform gray screen), 100% contrast sine-wave gratings moving in 16 directions, and plaids with seven different component intersection angles (dOri = 22.5, 45, 67.4, 90, 112.5, 135, and 157.5 deg) moving in 16 directions. In a subset of experiments, we also included 50% contrast gratings moving in 16 directions. Spatial frequency (0.05–0.15 cycles/deg), temporal frequency (3–6 Hz), and stimulus size (circular aperture of radius 10–35 deg) were optimized at each recording location. For each stimulus, the initial phase of each grating was chosen randomly from four possible values (0, 90, 180, and 270 deg). Thirty or 45 trials were run for each experiment, so that each stimulus was presented at least 10 times.

## Data analysis

### Direction selectivity and classic stimulus presentations for plaids

For direction selectivity measurements and experiments using the classic stimulus presentation mode (*Figure 1—figure supplement 1* and *Figure 2*), neuronal responses were calculated as the firing rate during stimulus presentation minus the firing rate during the last second of the pre-stimulus period. Tuning properties were then computed based on the mean responses across stimulus repetitions for each condition.

Direction selectivity was quantified using a direction index comparing responses between preferred and null directions, which was computed as follows:

$$DSI = 1 - \frac{R(N)}{R(P)}$$

where R(P) is the response to the preferred direction and R(N) is the response to the null direction. We also computed an orientation selectivity index, which was used for cell selection in the plaid analysis (see below). This index was computed as follows:

$$OSI = 1 - \frac{R(O)}{R(P)}$$

where R(P) is the response to the preferred direction and R(O) is the mean response to the two gratings with an orientation orthogonal to the preferred. Both indices are computed based on responses to gratings only.

To quantify motion integration in the classic paradigm, we used standard methods to compute partial correlations between the measured plaids responses and pattern and component response predictions (*Movshon et al., 1985*). Partial correlations were then Z-transformed to achieve a normal distribution that takes into account the degrees of freedom and better allows comparisons across conditions. The Z-transform was computed as (*Smith et al., 2005*):

$$Z = \sqrt{N-3}\,\frac{1}{2}\ln\left(\frac{1+r}{1-r}\right)$$

where r is the partial correlation (either pattern or component) and N refers to the number of points in the correlation (here, 16). Cells were classified as pattern cells if they met $Z_P - Z_c > 1.28$ for $Z_C \geq 0$, and $Z_P > 1.28$ otherwise. Component cells had to meet the opposite criterion. We also computed a pattern index as $Z_P – Z_C$. As in the cell classification, any negative values ($Z_P$ or $Z_C$) were set to 0 when computing the index.

## Streaming stimulus presentation

Stimulus-evoked responses collected using the streaming stimulus paradigm were extracted after computing an optimal latency for every neuron, as described in our previous work (*Lempel and Nielsen, 2019*). After computing the response to each condition, each neuron's pattern and component predictions were computed as described in *Lempel and Nielsen, 2019*. Briefly, we computed a dOri curve for every neuron as the average across all stimuli with the same dOri. We then combined the dOri curve with a direction tuning curve to generate a pattern and component prediction. For the pattern prediction, this direction tuning curve was computed by averaging responses to gratings and plaids with shared directions, using the pattern direction as the plaid's direction. For the component prediction, the direction tuning curve was instead computed based on the component directions (again, gratings were included in this computation). Finally, we computed partial correlations of each neuron's actual response with the two predictions and converted these values to Z-scores as before (with N set to 112 to account for the 7 dOri values and 16 directions involved in the computation).

## **Inclusion criteria and data set size**
### Direction selectivity and classic stimulus presentations for plaids

For direction selectivity measurements and experiments using the classic stimulus presentation mode (*Figure 1—figure supplement 1* and *Figure 2*), we first excluded neurons that did not meet a minimal response criterion by eliminating all neurons for which the best condition had an average firing rate of less than 2 Hz. The remaining neurons were then screened for general stimulus responsiveness. For direction selectivity measurements (*Figure 1—figure supplement 1*), we performed this test by using a one-way ANOVA to compare responses across all stimulus conditions (including the blank). Only cells that passed p<0.01 for the ANOVA were included in further analyses. For the plaid experiments (*Figure 2*), two responsiveness tests were performed: one for gratings and one for plaids only. Both tests used a one-way ANOVA to compare responses across all stimulus conditions plus the blank. Cells that passed both tests with p<0.05 were included in further analyses.

### Streaming stimulus presentation

Cells were screened for responsiveness using two criteria: First, cells had to pass an ANOVA comparing the responses for all plaid conditions with dOri = 90 deg and the blank condition (criterion p<0.01). Second, the responses to the best grating had to be larger than 2 Hz and responses to the best plaid with dOri = 90 deg had to be larger than 1.5 Hz. The criterion rate for the plaids was lowered slightly to account for generally lower firing rates for these conditions. We also applied a minimal tuning criterion because component and pattern predictions could not reasonably be computed for untuned neurons. To meet the tuning criterion, cells had to be orientation selective (OSI > 0.6) and direction selective (DSI > 0.7). Again, these indices were computed strictly based on responses to gratings, not plaids.

## Data set size

The number of animals and neurons for every figure is listed in *Table 2*. The uneven size of the data sets for different age groups is due to the fact that more experiments were run in younger animals (as part of other studies). Generally, slightly more neurons were excluded in younger than in older animals because responses in younger animals were weaker overall. For the main PSS data set for the streaming stimulus protocol, our selection criteria had the following consequences: At P37–70, 607 single units could be sorted. Of these, 458 (75%) were responsive to gratings, with 297 (49%) passing the direction and orientation criterion. Finally, 153 of these neurons (25% of total number) also passed the responsiveness criterion for plaids. At P41–47, 265 neurons were sorted, of which 215 (81%) were responsive to gratings, 147 (55%) were sufficiently orientation and direction selective, and 85 neurons (32%) also passed the responsiveness test for plaids. In adults, 100 neurons were sorted, of which 88 responded to gratings (88%), 63 (63%) were direction and orientation selective, and 46 (46%) were responsive to plaids.

## Statistics

Direction selectivity, pattern index, pattern and component Z-corrected correlations were compared between experimental groups using a Welch's t-test. This version of the Student's t-test is modified to accommodate unequal sampling and unequal variance across experimental groups. Model parameters determined through data fits were compared between age groups using the non-parametric Wilcoxon rank-sum test.

In cases of significant sample size differences between two groups (e.g., different age groups for PSS data and model fits), we performed an additional re-sampling test to verify significant differences in the median of a given metric (such as the pattern index or a model parameter). One thousand data sets were computed by randomly subsampling the larger of the two groups. In the random re-sampling procedure, N data points were randomly drawn (with replacement) from the larger data set, where N equals the number of samples in the smaller data set. For each of the 1000 data sets, we then computed the median and compared it to the median of the entire (larger) data set by computing their difference. This yielded a distribution of median differences for the 1000 randomly

**Table 2.** Number of animals and neurons for all experiments.

| Figures | Experiment/analysis | Experimental group | Animals | Neurons |
|---|---|---|---|---|
| 1 Sup 1 | Analysis of direction selectivity using gratings. | PSS. P37. | 3 | 37 |
| 1 Sup 1 | Analysis of direction selectivity using gratings. | PSS. Adult. | 10 | 68 |
| 1, 2, and 4 | Analysis of pattern index and dOri tuning using responses to plaids of different dOri values. | PSS. P37–40. | 27 | 153 |
| 1, 2, and 4 | Analysis of pattern index and dOri tuning using responses to plaids of different dOri values. | PSS. P41–47. | 13 | 84 |
| 1, 2, and 4 | Analysis of pattern index and dOri tuning using responses to plaids of different dOri values. | PSS. Adult. | 9 | 46 |
| 3 | Analysis of pattern index and dOri tuning using responses to plaids of different dOri values. | PSS. V4. | 4 | 33 |
| 3 | Analysis of pattern index and dOri tuning using responses to plaids of different dOri values. | PSS. V5. | 8 | 34 |
| 3 | Analysis of pattern index and dOri tuning using responses to plaids of different dOri values. | PSS. P37. | 6 | 36 |
| 3 | Analysis of pattern index and dOri tuning using responses to plaids of different dOri values. | PSS. P40. | 8 | 38 |
| 2 Sup 1 | Analysis of pattern index using responses to plaids of dOri 135 deg. | PSS. P37–40. | 6 | 12 |
| 2 Sup 1 | Analysis of pattern index using responses to plaids of dOri 135 deg. | PSS. P41–47. | 5 | 23 |
| 2 Sup 1 | Analysis of pattern index using responses to plaids of dOri 135 deg. | PSS. Adult. | 7 | 35 |
| 5, 6 | Analysis of pattern index and dOri tuning using responses to plaids of different dOri values. | V1. P37–40. | 16 | 97 |
| 5, 6 | Analysis of pattern index and dOri tuning using responses to plaids of different dOri values. | V1. P41-47. | 14 | 115 |
| 5 | Analysis of pattern index and dOri tuning using responses to plaids of different dOri values. | V1. Adult. | 9 | 40 |
| 6 | Analysis of pattern index and dOri tuning using responses to plaids of different dOri values. | Muscimol | 5 | 38 |
| 6 | Analysis of pattern index and dOri tuning using responses to plaids of different dOri values. | ACSF | 3 | 38 |
| 7, 8 | Analysis of model MLE fits. | PSS. P37-40. | 25 | 136 |
| 7 | Analysis of model MLE fits. | PSS. P44-47. | 8 | 46 |

subsampled data sets. Finally, the difference in medians between the two experimental groups was compared against the difference distribution determined by the re-sampling procedure to obtain the test's p-value.

To test for differences in relative plaid responses across dOri values, age groups, and their interaction, a two-way ANOVA with an interaction term was used. We found that the distribution of relative plaid responses had a long tail at the high end, typical of a $\chi^2$-distribution. To transform this distribution in a way that would better fit the assumptions of this parametric test, we performed a Wilson–Hilferty transformation (*Wilson and Hilferty, 1931*) of the data by computing the cubic root. This transformation largely removed the asymmetries in the distribution (*Figure 4—figure supplement 1*). This transformation was also used before computing correlations between relative plaid responses and pattern index (*Figure 4C*). For comparisons of Wilson–Hilferty transformed relative plaid responses at particular dOri values, a Welch's t-test was used.

## Image-computable motion-pathway model

The model used here is a modification of a model we described previously (*Lempel and Nielsen, 2019*). The code for the new model is available on github (https://github.com/nielsenlabmbi/PSSModelMLE; *Lempel, 2020*; copy archived at swh:1:rev:7a4bba49cd2c72-fab0623f3c3756fd55506cc816 *Lempel, 2021*). The LGN and V1 stages of the model described in *Lempel and Nielsen, 2019* were removed and instead responses of V1 cells providing input to PSS were modeled after empirical data collected from V1 neurons at different developmental stages. Model V1 responses were computed from empirical data in the following way: Data from individual neurons (as a function of plaid direction and dOri) were first normalized by their maximum response to gratings and shifted in direction space to align the preferred direction to 0 deg. Data from neurons belonging to the same age group were then averaged to compute a single V1 filter profile per group. This profile was used to generate 16 V1 filters with different direction preferences (homogeneously covering direction space) by shifting the profile in direction space.

In the previous implementation of this model (*Lempel and Nielsen, 2019*), V1 response profiles were completely direction selective and symmetric around the preferred direction, in that stimuli moving in directions equidistant to the preferred direction elicited identical responses. To maintain these two properties of V1 filters in the new model implementation, we applied the following two transformations: To make the filters symmetric, we effectively mirrored the 2D response profile for each filter around the preferred direction. More precisely, at each dOri the filter response for directions with similar distances from the preferred direction were averaged. The original responses were then replaced with the average. To make the output of the V1 stage perfectly direction selective, we combined the responses of V1 model neurons with opposite direction preferences. More precisely, each direction channel consisted of a pair of V1 filters. The response of the filter was computed as P-P(o)O, where P is the response of filter one at a particular direction and dOri, O the response of filter two which prefers the opposite direction, and P(o) the response of filter one to a grating moving in the null direction. This forced the response of the filter pair to be 0 at the null direction. Any negative values resulting from the computation were set to 0.

The V1 filter stage was followed by a PSS integration stage as described previously. Briefly, the responses of the 16 V1 filters were combined linearly, using a weight function that consisted of two van Mises functions, one centered at the preferred and one centered at the null direction. This PSS integration stage was followed by a final non-linearity, implemented here as an exponential function (as opposed to the subtraction and rectification method implemented previously) so that:

$$PSS_{resp\ after\ NL} = \left(PSS_{resp}\right)^{T_{PSS}}$$

This change was implemented to aid the parameter optimization method described below. In total, the model had the following free parameters: $K_E$, $K_I$, I, $T_{PSS}$, and $Resp_{max}$. This last variable simply multiplied PSS responses computed by the model to best match empirically recorded firing rates.

## Model fitting

Model parameters were fit to maximize the likelihood of responses measured empirically from individual PSS neurons. V1 responses used in the model were selected to match the age group of the

empirical PSS data being fitted. Likelihood of empirical data was computed from modeled parameters assuming a Poisson distribution with event frequency equal to the PSS response computed by the model. Log-likelihood was computed for the empirically measured response of the PSS neuron in every trial across all stimuli conditions and then summed to compute the log-likelihood of all observed data given certain model parameters. Model parameters were optimized to maximize log-likelihood using a simulated annealing algorithm implemented by the Matlab function 'smulannealbnd' (Global Optimization Toolbox). Lower and upper bounds for variables were as follows: $K_E$ [−5–6], $K_I$ [−5–6], I [0–2], $T_{PSS}$[0–3], and $Resp_{max}$ [80% max. response of empirical data – 120% max. response of empirical data]. As a test of fit quality for the model, we computed the Pearson correlation between mean empirical responses and modeled responses across stimuli. Model fits with correlation coefficients below 0.5 were removed from further analysis. Seventeen of 153 model fits in the P37–40 age group, and 6 of 52 model fits in the P44–47 age group were removed in this manner.

## Acknowledgements

We are grateful to J Killebrew, W Nash, and W Quinlan for technical support and R Krauzlis for advice on the muscimol injections. We thank R Srinath, E Dunn-Weiss, A Emonds, and other members of the Nielsen and Connor labs for helpful discussions and experimental support.

## Additional information

### Funding

| Funder | Grant reference number | Author |
|---|---|---|
| National Eye Institute | EY027853 | Augusto A Lempel<br>Kristina J Nielsen |

The funders had no role in study design, data collection and interpretation, or the decision to submit the work for publication.

### Author contributions

Augusto A Lempel, Conceptualization, Data curation, Software, Formal analysis, Validation, Investigation, Visualization, Methodology, Writing - original draft, Project administration, Writing - review and editing; Kristina J Nielsen, Conceptualization, Supervision, Funding acquisition, Methodology, Writing - original draft, Project administration, Writing - review and editing

### Author ORCIDs

Augusto A Lempel ![ORCID] https://orcid.org/0000-0002-3644-2690
Kristina J Nielsen ![ORCID] https://orcid.org/0000-0002-9155-2972

### Ethics

Animal experimentation: All procedures were performed in strict accordance with the guidelines of the National Institute of Health and were approved by the Animal Care and Use Committee of Johns Hopkins University (Protocol Numbers FE15M305 and FE18M239).

### Decision letter and Author response

Decision letter https://doi.org/10.7554/eLife.59798.sa1
Author response https://doi.org/10.7554/eLife.59798.sa2

## Additional files

### Supplementary files

• Transparent reporting form

## Data availability

Source code for the computation model is provided as a github repository.

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
