## [Decision Letter]

**Acceptance summary:**

This work provides important new insights into the development of visual motion integration in the ferret visual system. By simultaneously following the development of responses to global motion in primary visual cortex (V1) and in the higher order cortical area PSS, the paper demonstrates that responses to complex motion in the two areas depend on mutual and coordinated interactions, documenting the prominent role of feedback during development. The work has important implications for broader understanding of the nature of interactions between processing stages during brain development.

**Decision letter after peer review:**

Thank you for submitting your article "Development of visual motion integration involves coordination of multiple cortical stages" for consideration by *eLife*. Your article has been reviewed by two peer reviewers, and the evaluation has been overseen by a Reviewing Editor and Chris Baker as the Senior Editor. The following individual involved in review of your submission has agreed to reveal their identity: Robbe Goris (Reviewer #1).

The reviewers have discussed the reviews with one another and the Reviewing Editor has drafted this decision to help you prepare a revised submission.

As the editors have judged that your manuscript is of interest, but as described below that additional analyses are required before it is published, we would like to draw your attention to changes in our revision policy that we have made in response to COVID-19 (https://elifesciences.org/articles/57162). First, because many researchers have temporarily lost access to the labs, we will give authors as much time as they need to submit revised manuscripts. We are also offering, if you choose, to post the manuscript to bioRxiv (if it is not already there) along with this decision letter and a formal designation that the manuscript is "in revision at *eLife*". Please let us know if you would like to pursue this option. (If your work is more suitable for medRxiv, you will need to post the preprint yourself, as the mechanisms for us to do so are still in development.)

Summary:

Both reviewers were impressed with several aspects of the study and felt that it has a potential of providing new insights into the development of mechanisms underlying cortical processing of global motion. However, they raised a number of substantive concerns that must be addressed before the manuscript can be considered for publication in *eLife*. These concerns are summarized below.

1) The criteria for establishing developmental epochs are inconsistent throughout the manuscript. These should be established ahead time, justified and used consistently for all analyses.

2) Please address the issues with data analysis resulting from inconsistent subsampling of the data, exemplified in Figures 2 and 5 (see comments from reviewer 2)

3) A related problem results from potential sampling bias: firm conclusions are drawn about group differences despite large differences in Ns among the groups. Adding information about how representative the samples are and a more robust analysis approach could allay these concerns.

4) Please address the problem with the current version of the model-based analysis, raised by Reviewer 1. This reviewer suggests fitting the data with the maximum likelihood method and made a number of other suggestions that are likely to improve the analysis and fitting the data.

You may want to consider this reviewer's suggestion to reorganize the paper and start with presenting the data followed by the introduction of the model and its simulations.

5) Please address the issue raised by reviewer 1 concerning the discussion of the results starting on paragraph four of the Results section.

6) Each reviewer provided additional comments and suggestions, highlighting a number of inconsistencies, which I suggest you address directly.

Reviewer #1:

Lempel and Nielsen report a very cool empirical result: the coordinated development of visual motion integration in areas V1 and PSS. The insight that feedback plays a more prominent role in specific phases of visual development is novel and important. That said, the paper offers no helpful computational understanding of this phenomenon, is written in a manner that appears to reflect the "autobiographical" logic of studies more so than what is actually learned from these studies, and contains a number of questionable data-analysis choices that should either be improved or better justified.

Summary of substantive concerns

1) The age-grouping differs throughout the paper. This is very problematic for a developmental study. The authors need to establish the developmental epochs of interest and the criteria to group ages up front and then stick to these conventions throughout the paper.

2) The model-based analysis in its current form is not helpful. There are a number of issues. First, when introducing the model, it would be helpful if you would start by contrasting the model-parameters for the component and pattern cells – is there any meaningful difference? This is not obvious. For example, the width of the excitatory channel appears to be going in the "wrong" direction in light of the root model of Simoncelli and Heeger (1998) – pattern cells ought to have a broader direction of motion bandwidth, see Figure 10 in Zaharia et al., 2019, for empirical evidence in primates. Second, a recovery analysis would build trust in the modeling enterprise and reveal whether this set of empirical measurements actually allows to unambiguously identify the model's parameters (this could go in supplementary information). Third, the use of mean square error as badness-of-fit statistic and grid-search as optimization routine likely introduces a needless amount of bias and variability in the parameter estimates. How about fitting the data using a maximum likelihood method and an explicit model of spike generation, for example a modulated Poisson process as in Zaharia et al. (2010)? That approach better reflects the standard of model-based data-analysis in current systems neuroscience. Fourth, how can you describe the plots in Figure 5C and D as a match? The model obviously fails to reproduce the most prominent features of the data, calling into question whether we can learn anything meaningful from its parameter estimates at all. Models are only useful if they create insight beyond what can be learned from a simpler data-analysis. It's not clear that the current analysis meets that standard.

3) I would consider to reorganize the paper, lead with the cool coordinated development result, and only then turn to the question "Which computational mechanisms might underlie this phenomenon". I would be fine with a model simulation rather than a model-fitting exercise if it provided some insight into how this coordination can come about. I think that a more complex two-stage model with a feedback loop would be appropriate. How would the feedforward and feedback elements of such a model need to change to capture your empirical observations in both areas? Perhaps you can derive an empirically testable prediction from such a simulation?

4) Results paragraph four and following. The developmental changes in PSS neurons' motion integration are described and treated as a number of independent effects. "First, the influence of the component signals decreased with age… Second, the influence of the pattern signal increased with age…" etc. This is based on an analysis in which data are correlated with two competing predictions. Due to the nature of correlation, it typically has to be the case that if one number goes up, the other has to come down. I would consider reducing this to a single clear statement: The fraction of pattern cells changes, as does the average pattern-index of the population.

Reviewer #2:

This paper describes a set of experiments designed to address the question of how global motion sensitivity develops in ferret high order cortical area PSS. Sensitivity to grating and a range of plaid stimuli is tested in ferrets of different – but early – ages to address this question. The findings can apply to questions of brain development more broadly, as they point to a hierarchical developmental profile – one for which there is fairly strong support in the broader development literature. Thus, the data are of value beyond the specific focus of this paper – on development of motion mechanisms in ferret. Strengths include the comparative analysis of multiple processing stages over the same developmental time period, comparison of low-level and higher-level motion mechanisms, and comparison of their data with a model that allows one to draw insights into the neural mechanisms that contribute to the development of plaid motion sensitivity. The main weaknesses are related to a general sense of uneasiness about the data analysis and presentation, that leaves the reader less than convinced about some of the conclusions.

Strengths

Recordings are made from V1 and PSS, although not in the same animals. Simultaneous recordings would be preferred because of interanimal variability but that was not done here. Regardless, some conclusions can be drawn.

Ferret model allows access to a very early time period in visual system development that is not accessible in the primate. Also, ferret has an analogous motion processing hierarchy to the primate suggesting that the data are likely to apply generally.

The authors utilize a modified version of a previously developed model for plaid sensitivity. Direct comparison is made between data and model predictions, with the goal of identifying potential mechanisms driving the developmental changes.

Weaknesses

Throughout the manuscript, different comparisons are made using different poolings of the neural data. It is unclear why this is. It seems as though they have a large data set collected from many ferrets of a range of ages that is then selectively subsampled for different comparisons. This is not a robust or reliable approach and seems arbitrary and ad hoc. For the primary comparisons, Figure 2, 4 groups of animals are described (P37-41, P42-43, P44-47, "adult"). For the model, 2 groups are selected, here – justifiably – to subdivide before by and after Plaid sensitivity development: P37-41, P44->adult. However, in Figure 5, the data are divided into 3 groups: P37-41, P44-47or 48, "adult". Apart from the case of the model, there is no principled reason for changing the age groupings for the animals.

Perhaps related to the above concern, conclusions are drawn with certainty based on often lopsided comparisons across groups with grossly unequal N. For example, the fundamental conclusion regarding the age at which the pattern response matures is based on a small number of observations in the critical age group P42-43 (N=31); the youngest age group has N=154 while the older age groups have moderate although still comparatively small N. According to Table 2, those 31 neurons came from 5 animals. Similarly, some of the supplemental figures are based on very small numbers of units (e.g. S3c, N=11). One has to wonder about sampling bias and individual differences across animals. Are these differences really meaningful? What is the relative proportion of Zc, Zp and pattern index across different animals of the same age/experience? Without this variance information it is hard to know if the conclusions are valid.

Similarly, although the analysis of the effect of visual experience is very clever, one has to wonder why the data are not presented by days. V4 – a single day – group is compared with "5 or more days" range 5-7 days, resulting in a comparison between N=33 and N=105 units. There is no principled reason for this choice given. For this comparison to be meaningful, the individual days with (presumably) more comparable N should be presented.

The muscimol experiment seems to be tacked on the end. It is a critical component of the overall study that shows that there is likely an interaction between the maturation of PSS and V1 plaid responses. The earlier conclusion based on the model that V1 responses change and play a role in maturation of motion responses, could be because of feedback. These points are not connected until later. Better to include them earlier and provide a more coherent narrative.

The main analysis would be more robust if based on a coherent multivariate analysis that accounts for multiple comparisons to any one data set.

[Editors' note: further revisions were suggested prior to acceptance, as described below.]

Thank you for submitting your article "Development of visual motion integration involves coordination of multiple cortical stages" for consideration by *eLife*. Your article has been reviewed by two peer reviewers, and the evaluation has been overseen by a Reviewing Editor and Chris Baker as the Senior Editor. The following individuals involved in review of your submission have agreed to reveal their identity: Robbe Goris (Reviewer #1); Lynn Kiorpes (Reviewer #2).

Essential revisions:

The reviewers felt that most of the reservations raised in their reviews have been addressed. However, one last issue remains. It concerns the approach chosen to isolate the role of visual experience. The suggestion is to include all the data in the analysis, rather than just the subset. Specifically, it would be more informative to include a plot (similar to Figure 2A) showing each animal's median pattern index a function of visual experience over a reasonable range (e.g., from V1 to V7+ in steps of 1 day). The reviewers agreed that an ANCOVA analysis would be an appropriate statistical test for the hypothesis that visual experience explains variance in pattern index beyond the effects of gestational age.

Reviewer #1:

The authors addressed most of my concerns. The remaining issues are detailed below.

The paper has improved substantially, and the authors have addressed several of my concerns. One substantial issue remains in my opinion. Just like the grouping in developmental epochs felt a bit ad hoc in the previous submission, so does the grouping in visual-experience epochs in this resubmission. It is not clear what makes V4 and V5 so special that these epochs should be at the center of this analysis. I am not sure that the authors have sufficient data to distinguish experience from gestational age. The gestational age difference between the V4 and V5 group is a bit more than a day, and so seemingly highly correlated with the difference in visual experience. Furthermore, I don't understand the logic behind Figure 3D and 3E: Why is the absence of an age-based difference in pattern-selectivity evidence in favor of the role of visual experience? If the authors wish to attempt to isolate the role of visual experience, it would be better to conduct an analysis that involves all of their data, rather than just this subset. For example, a plot like Figure 2A, whereby each animal's median pattern index is shown as a function of visual experience over a reasonable range (e.g., from V1 to V7+ in steps of 1 day) would be more informative. I think that an ANCOVA analysis provides a suitable statistical test for the hypothesis that visual experience explains variance in pattern index beyond the effects of gestational age (Pattern index of each cell/animal: dependent variable; age group: independent variable; visual experience: covariate).

Reviewer #2:

The authors have substantially revised this manuscript. They have addressed the major issues raised in the reviews and followed the recommendations made. I find the data presentation to be clear and convincing, and the conclusions to be appropriate. My concerns have been addressed.

---

## [Author Response]

Summary:Both reviewers were impressed with several aspects of the study and felt that it has a potential of providing new insights into the development of mechanisms underlying cortical processing of global motion. However, they raised a number of substantive concerns that must be addressed before the manuscript can be considered for publication in eLife. These concerns are summarized below.1) The criteria for establishing developmental epochs are inconsistent throughout the manuscript. These should be established ahead time, justified and used consistently for all analyses.

We apologize for the changing epochs in the previous version of the manuscript, and have implemented consistent groups in the revised manuscript. To explain how these groups were selected, we now include a new plot in Figure 2 (Figure 2A) which shows the development of motion integration by plotting the median pattern index per animal as a function of age. All animals that were used in the study are represented in this figure to give a complete overview over the data set. Based on this plot, we divided the PSS data into 3 age groups: P37-40, P41-47 and adult. We explain this choice in the first subsection of the Results:

“While this plot confirmed that a range of pattern indices could be observed at every age, it also showed that the lowest median pattern indices generally occurred in animals before P41. After P41, the range of median pattern indices across animals of the same age was generally similar to that of adults. We therefore divided all developmental data into two groups, P37-40 and P41-47. A third group consisted of adult animals (age range P100-448).”

The same 3 groups are consistently used throughout the revision for all parts that concern PSS. Our V1 data set comes from a slightly narrower age range for the second group, P44-47. We therefore limit the PSS data set to the same age range when combining V1 and PSS data in the computational model. This is explained as well in the Results:

“We selected two age groups for these experiments. The first covered the same age range as the youngest PSS animals, P37-40. The second group covered P44-47, chosen to coincide with the end of the age range of the second PSS group so as to maximize our ability to detect age-dependent differences.”

And “Note that because the V1 data only covered P44-47, we restricted the PSS data set to the same range for the modeling analysis.”

2) Please address the issues with data analysis resulting from inconsistent subsampling of the data, exemplified in Figures 2 and 5 (see comments from reviewer 2)

As explained in the answer to item 1, we now only use 3 age groups throughout the manuscript. We also have moved the description of the cell selection criteria into a separate section in the Materials and methods, and – in addition to listing the number of animals and neurons per figure in Table 2 – list how the cell selection criteria impact our main PSS data set in that Materials and methods section. Regarding this latter point, the Materials and methods section now contains the following paragraph:

“Data set size: The number of animals and neurons for every figure are listed in Table 2. The uneven size of the data sets for different age groups is due to the fact that more experiments were run in younger animals (as part of other studies). Generally, slightly more neurons were excluded in younger than in older animals because responses in younger animals were weaker overall. For the main PSS data set for the streaming stimulus protocol, our selection criteria had the following consequences: At P37-70, 607 single units could be sorted. Of these, 458 (75%) were responsive to gratings, with 297 (49%) passing the direction and orientation criterion. Finally, 153 of these neurons (25% of total number) also passed the responsiveness criterion for plaids. At P41-47, 265 neurons were sorted, of which 215 (81%) were responsive to gratings, 147 (55%) were sufficiently orientation and direction selective, and 85 neurons (32%) also passed the responsiveness test for plaids. In adults, 100 neurons were sorted, of which 88 responded to gratings (88%), 63 (63%) were direction and orientation selective, and 46 (46%) were responsive to plaids.”

Numbers remain uneven between the different age groups. This is a consequence of larger number of experiments in the youngest animals. The data set here combines data collected as part of multiple studies for this age group, which yielded a larger N than for the other groups. We have adjusted all statistics to be appropriate for uneven sample sizes, and explicitly discuss the choice of statistics as part of the Materials and methods section as well.

3) A related problem results from potential sampling bias: firm conclusions are drawn about group differences despite large differences in Ns among the groups. Adding information about how representative the samples are and a more robust analysis approach could allay these concerns.

We have tried to address this concern in a number of ways. In particular, we now discuss how we take uneven sampling numbers into account in the statistical tests. We are using a Welch’s t-test instead of a Student’s t-test because of the uneven sample sizes, and are further confirming differences through a re-sampling procedure. Where necessary, we also first transform the data to achieve normal distributions before performing statistical tests (this occurred for the relative plaid responses). All tests and their results are listed in Table 1. Regarding the choice of statistics, the Materials and methods section now states:

“Direction selectivity, pattern index, pattern and component Z-corrected correlations were compared between experimental groups using a Welch’s t-test. This version of the Student’s t-test is modified to accommodate unequal sampling and unequal variance across experimental groups. Model parameters determined through data fits were compared between age groups using the non-parametric Wilcoxon rank-sum test.

In cases of significant sample size differences between two groups (e.g., different age groups for PSS data and model fits) we performed an additional re-sampling test to verify significant differences in the median of a given metric (such as the pattern index or a model parameter). 1000 data sets were computed by randomly subsampling the larger of the two groups. In the random re-sampling procedure, N data points were randomly drawn (with replacement) from the larger data set, where N equals the number of samples in the smaller data set. The metric’s median was computed for each of the 1000 data sets and compared to the median of the entire larger data set by computing the difference between the two medians. This yielded a distribution of the median differences for the 1000 randomly subsampled data sets. Finally, the difference in median between the two experimental groups was compared against the difference distribution determined by the re-sampling procedure to obtain the test’s p-value.”

Reviewer 2 also raised the concern that inter-animal variability might drive our findings. To address this concern, Figure 2A plots our data on a per-animal basis, which allows a full assessment of the developmental time course. Differences between younger and older animals are also confirmed by a significant difference in the median pattern index, computed per animal, across our 3 age groups (in addition to the differences when pooling pattern indices across neurons, independent of the animal they were collected in).

4) Please address the problem with the current version of the model-based analysis, raised by Reviewer 1. This reviewer suggests fitting the data with the maximum likelihood method and made a number of other suggestions that are likely to improve the analysis and fitting the data.You may want to consider this reviewer's suggestion to reorganize the paper and start with presenting the data followed by the introduction of the model and its simulations.

We have followed the reviewer’s suggestions. The paper now describes all of the experimental data first, before using the model to identify possible contributions made by the development of PSS-internal mechanisms and the observed change in V1 responses. We have generally modified the model so that the initial stage can be set to match our empirical measurements. As part of these changes, the model now also uses a maximum likelihood method to fit the data. The changes to the model are further detailed in the response to reviewer 1 below.

5) Please address the issue raised by reviewer 1 concerning the discussion of the results starting on paragraph four of the Results section.

We have followed the reviewer’s advice on this issue. We now highlight the change in pattern index and pattern cell proportion in describing the development. We still discuss the changes in pattern and component correlation for completeness sake, since they represent the data underlying the pattern index. We generally agree with the reviewer’s concern that these measures are based on correlating the same data with 2 predictions, and that the two partial correlations cannot be considered as truly independent. However, the demonstration that the two correlation coefficients have different latencies after stimulus onset (as shown in Smith, Majaj and Movshon 2005) shows that sometimes interesting differences can emerge between them. This is another reason for reporting the development of pattern and component correlations.

The relevant text in the result section now reads: “Consistent with the trends shown in Figure 2A, the pattern index increased significantly between P37-40 and P41-47, at which point it became adult-like (P37-40 vs P41-47: p<.001. See Table 1 for the other p values. Non-significant results are omitted from the results text, but are listed in Table 1). This change in pattern index was driven by opposite changes in both of the underlying partial correlation coefficients (Figure 2—figure supplement 1): Between the two younger age groups, Z_C_ decreased significantly (p<.001), while Z_P_ increased (p<.001). Changes in pattern and component cell proportions show the same trends (Figure 2C): The proportion of component cells decreased with age (75% at P37-40 vs 41% in adults), while the proportion of pattern cells increased (13% at P37-40 vs 41% in adults).”

6) Each reviewer provided additional comments and suggestions, highlighting a number of inconsistencies, which I suggest you address directly.Reviewer #1:Lempel and Nielsen report a very cool empirical result: the coordinated development of visual motion integration in areas V1 and PSS. The insight that feedback plays a more prominent role in specific phases of visual development is novel and important. That said, the paper offers no helpful computational understanding of this phenomenon, is written in a manner that appears to reflect the "autobiographical" logic of studies more so than what is actually learned from these studies, and contains a number of questionable data-analysis choices that should either be improved or better justified.Summary of substantive concerns1) The age-grouping differs throughout the paper. This is very problematic for a developmental study. The authors need to establish the developmental epochs of interest and the criteria to group ages up front and then stick to these conventions throughout the paper.

As outlined above, the paper now first shows how pattern integration develops as a function of age across the entire data set. Groups are then established, and then same groups used throughout.

2) The model-based analysis in its current form is not helpful. There are a number of issues. First, when introducing the model, it would be helpful if you would start by contrasting the model-parameters for the component and pattern cells – is there any meaningful difference? This is not obvious. For example, the width of the excitatory channel appears to be going in the "wrong" direction in light of the root model of Simoncelli and Heeger (1998) – pattern cells ought to have a broader direction of motion bandwidth, see Figure 10 in Zaharia et al., 2019, for empirical evidence in primates. Second, a recovery analysis would build trust in the modeling enterprise and reveal whether this set of empirical measurements actually allows to unambiguously identify the model's parameters (this could go in supplementary information). Third, the use of mean square error as badness-of-fit statistic and grid-search as optimization routine likely introduces a needless amount of bias and variability in the parameter estimates. How about fitting the data using a maximum likelihood method and an explicit model of spike generation, for example a modulated Poisson process as in Zaharia et al. (2010)? That approach better reflects the standard of model-based data-analysis in current systems neuroscience. Fourth, how can you describe the plots in Figure 5C and D as a match? The model obviously fails to reproduce the most prominent features of the data, calling into question whether we can learn anything meaningful from its parameter estimates at all. Models are only useful if they create insight beyond what can be learned from a simpler data-analysis. It's not clear that the current analysis meets that standard.

We have changed the model approach in a number of ways. Conceptually, the model is now used to investigate how the developmental change seen between the 2 younger groups might be accomplished by the most likely candidate mechanisms (we no longer use the model for the adult data in the current manuscript). The candidate mechanisms considered here are changes in V1 input, changes in PSS null direction inhibition and PSS excitatory interactions between the component signals. To be able to look at the contribution of each of these mechanisms, we have revised the model. It still consists of 2 stages linked in a strictly feedforward manner (see answer to point 3 below as well). However, the first V1-like stage is no longer determined through fitting the model to PSS data. Instead, we set it to match the empirical data at a particular age. It is followed by a PSS-like stage that integrates the signals from different V1 direction filters. We first use this model to determine PSS changes between the 2 age groups. To this end, we use empirical V1 data from either P37-40 or P44-47 to set the first stage, and then fit the PSS stage to data from the matching age range. This returns significant changes in PSS inhibition and moderate changes in excitation. Obviously, V1 changes and PSS changes all work in concert (at least in the model). We still wanted to know what their individual contributions are to the developmental change in motion integration (or phrased differently, how they complement each other to achieve higher motion integration levels). The idea behind this effort was to gain some insight into whether the V1 changes actually could make a meaningful contribution in PSS. In this second part of the model exercise, we first fixed the V1 stage to its P37-40 state, and then either added more PSS inhibition, or more PSS excitation. Lastly, we kept PSS fixed at P37-40 and changed the V1 stage to its P44-47 setting. Note that no fitting was performed in this part, we simply adjusted the parameters of model neurons and studied the impact on their responses. We feel that this analysis contributed important insights: It shows that increased inhibition can be a major driver of increased pattern indices by suppressing component responses. At the same time, the increased inhibition causes a more widespread reduction in plaid responses, including to plaids with pattern motion in the preferred direction. PSS excitation changes and V1 changes work in concert to boost the responses to these conditions. The empirically observed V1 changes would in particular provide a boost to plaids with intermediate dOri values. We propose that this is necessary because there is a limit to how much PSS excitation can be enhanced without losing direction selectivity (excitation here refers to the width of the excitatory component of the integration function, i.e. what range of component signals is integrated across by the PSS neuron). All of these ideas are discussed in the Results section (in: Modeling the relative impact of developmental changes in V1 versus PSS on motion integration, Figures 7 and 8) and the Discussion.

3) I would consider to reorganize the paper, lead with the cool coordinated development result, and only then turn to the question "Which computational mechanisms might underlie this phenomenon". I would be fine with a model simulation rather than a model-fitting exercise if it provided some insight into how this coordination can come about. I think that a more complex two-stage model with a feedback loop would be appropriate. How would the feedforward and feedback elements of such a model need to change to capture your empirical observations in both areas? Perhaps you can derive an empirically testable prediction from such a simulation?

We thank the reviewer for this suggestion. We have reorganized the paper as suggested – the data is discussed first before describing the model. As outlined above, we now use the model to test which contributions the developmental changes in V1 and PSS make. We agree with the reviewer that in light of the inactivation experiments a more complex model incorporating feedforward and feedback mechanisms would be very interesting. At this point, the “standard” motion pathway models (including the one we adapted for the ferret in our previous publication) are strictly feedforward. Developing a more complex model is absolutely worthwhile, but at this point exceeded what we felt was possible for a revision. For now, the directly testable conclusions of the model are therefore limited to predictions regarding the developmental time course of inhibition and excitation in PSS, which we describe in the Discussion:

“Inhibition proved to be the strongest contributor to improving motion integration. Increasing inhibitory levels from those at P37-40 to the levels at P44-47 strongly increased the pattern index across the entire model neuron population. Our model therefore makes the testable prediction that inhibitory circuits in PSS should mature relatively slowly, not before P40. No data exist for inhibitory circuit development in PSS, but inhibitory circuits in ferret V1 have been found to develop up to P60 both in terms of the relative proportion of different types of inhibitory neurons, as well as their laminar distribution ^30–33^. A similarly slow development of inhibitory circuity in PSS does therefore appear possible.”

We also mention the idea of a more complex model in the Results section:

“Note that one major purpose of the model was to test whether the observed V1 changes could propagate to PSS (and how they would affect PSS responses). For this reason, the model remained strictly feedforward, despite the above observations that V1 changes depended on feedback from PSS. Future models will need to expand to more complicated models including both feedforward and feedback links to fully explore the dependencies between areas.“

4) Results paragraph four and following. The developmental changes in PSS neurons' motion integration are described and treated as a number of independent effects. "First, the influence of the component signals decreased with age… Second, the influence of the pattern signal increased with age…" etc. This is based on an analysis in which data are correlated with two competing predictions. Due to the nature of correlation, it typically has to be the case that if one number goes up, the other has to come down. I would consider reducing this to a single clear statement: The fraction of pattern cells changes, as does the average pattern-index of the population.

As described above, we have revised how the changes in pattern index and component and pattern correlation are discussed in the text to address this concern.

Reviewer #2:This paper describes a set of experiments designed to address the question of how global motion sensitivity develops in ferret high order cortical area PSS. Sensitivity to grating and a range of plaid stimuli is tested in ferrets of different – but early – ages to address this question. The findings can apply to questions of brain development more broadly, as they point to a hierarchical developmental profile – one for which there is fairly strong support in the broader development literature. Thus, the data are of value beyond the specific focus of this paper – on development of motion mechanisms in ferret. Strengths include the comparative analysis of multiple processing stages over the same developmental time period, comparison of low-level and higher-level motion mechanisms, and comparison of their data with a model that allows one to draw insights into the neural mechanisms that contribute to the development of plaid motion sensitivity. The main weaknesses are related to a general sense of uneasiness about the data analysis and presentation, that leaves the reader less than convinced about some of the conclusions.StrengthsRecordings are made from V1 and PSS, although not in the same animals. Simultaneous recordings would be preferred because of interanimal variability but that was not done here. Regardless, some conclusions can be drawn.Ferret model allows access to a very early time period in visual system development that is not accessible in the primate. Also, ferret has an analogous motion processing hierarchy to the primate suggesting that the data are likely to apply generally.The authors utilize a modified version of a previously developed model for plaid sensitivity. Direct comparison is made between data and model predictions, with the goal of identifying potential mechanisms driving the developmental changes.WeaknessesThroughout the manuscript, different comparisons are made using different poolings of the neural data. It is unclear why this is. It seems as though they have a large data set collected from many ferrets of a range of ages that is then selectively subsampled for different comparisons. This is not a robust or reliable approach and seems arbitrary and ad hoc. For the primary comparisons, Figure 2, 4 groups of animals are described (P37-41, P42-43, P44-47, "adult"). For the model, 2 groups are selected, here – justifiably – to subdivide before by and after Plaid sensitivity development: P37-41, P44->adult. However, in Figure 5, the data are divided into 3 groups: P37-41, P44-47or 48, "adult". Apart from the case of the model, there is no principled reason for changing the age groupings for the animals.

We hope to have eliminated these concerns by subdividing animals into 3 groups only, P37-40, P41-47 and adults as described above. The middle age group is narrowed further to P44-47 for the V1 experiments, but the limits delineating the different groups no longer change throughout the manuscript. We have added Figure 2A to show the full data set and explain the choice of age groups.

Perhaps related to the above concern, conclusions are drawn with certainty based on often lopsided comparisons across groups with grossly unequal N. For example, the fundamental conclusion regarding the age at which the pattern response matures is based on a small number of observations in the critical age group P42-43 (N=31); the youngest age group has N=154 while the older age groups have moderate although still comparatively small N. According to Table 2, those 31 neurons came from 5 animals. Similarly, some of the supplemental figures are based on very small numbers of units (e.g. S3c, N=11). One has to wonder about sampling bias and individual differences across animals. Are these differences really meaningful? What is the relative proportion of Zc, Zp and pattern index across different animals of the same age/experience? Without this variance information it is hard to know if the conclusions are valid.

As outlined above, we have adjusted all statistical tests to be more robust to uneven sample sizes. Our choices are discussed in the Materials and methods section. The reviewer correctly points out that there are inter-animal differences, in particular during development. For this reason, we now show the PSS pattern indices on a per-animal basis in Figure 2A, as requested by the reviewer. We hope the reviewer agrees with us that there are noticeable differences in pattern index between the youngest and older animals.

Similarly, although the analysis of the effect of visual experience is very clever, one has to wonder why the data are not presented by days. V4 – a single day – group is compared with "5 or more days" range 5-7 days, resulting in a comparison between N=33 and N=105 units. There is no principled reason for this choice given. For this comparison to be meaningful, the individual days with (presumably) more comparable N should be presented.

We thank the reviewer for this suggestion. We have revised the analysis exactly as suggested, and now compare the data for animals with 4 days of visual experience with that of animals with 5 days of visual experience. The N is now matched between groups (N=33 for the V4 group, N=34 for the V5 group, see Figure 3).

The muscimol experiment seems to be tacked on the end. It is a critical component of the overall study that shows that there is likely an interaction between the maturation of PSS and V1 plaid responses. The earlier conclusion based on the model that V1 responses change and play a role in maturation of motion responses, could be because of feedback. These points are not connected until later. Better to include them earlier and provide a more coherent narrative.

We agree about the importance of this experiment. It is now reported earlier in the manuscript, directly after describing the V1 recording results, and before discussing the model (the data are presented in Figure 6).

The main analysis would be more robust if based on a coherent multivariate analysis that accounts for multiple comparisons to any one data set.

We use multivariate analyses to compare effects of dOri and age on the relative plaid responses. We have not implemented it for the other comparisons, but hope that the reviewer finds the adjustments in the statistics sufficient to make our conclusions robust.

[Editors' note: further revisions were suggested prior to acceptance, as described below.]

Essential revisions:The reviewers felt that most of the reservations raised in their reviews have been addressed. However, one last issue remains. It concerns the approach chosen to isolate the role of visual experience. The suggestion is to include all the data in the analysis, rather than just the subset. Specifically, it would be more informative to include a plot (similar to Figure 2A) showing each animal's median pattern index a function of visual experience over a reasonable range (e.g., from V1 to V7+ in steps of 1 day). The reviewers agreed that an ANCOVA analysis would be an appropriate statistical test for the hypothesis that visual experience explains variance in pattern index beyond the effects of gestational age.

The youngest age included in this study was P37, and ferrets tend to open their eyes around P30. As a consequence, there is a lower limit on the amount of visual experience in the current data set – no animal had less than 4 days of visual experience. Because we limited the analysis to animals between P37 and 40 (which had the most immature levels of motion integration, and therefore would be the most sensitive group to test for an impact of visual experience), there also is an upper limit on the amount of visual experience. As outlined in the manuscript, we only encountered one animal with 7 days of visual experience in this group; the rest of the animals fell between 4 and 6 days of visual experience. We excluded the animal with 7 days of visual experience because it was such an outlier compared to the rest. As requested, we have included a plot similar to Figure 2A in Figure 3, which shows each animal’s median pattern index as a function of visual experience (Figure 3A). This plots shows a change in pattern index between 4 and 5 days of visual experience, after which the pattern index appears to remain more stable. We therefore grouped the data into 2 groups, one containing the animals with 4 days of visual experience, and the other group containing animals with 5 or 6 days of visual experience. The remainder of Figure 3 then tests differences in plaid responses between these groups.

We appreciate the request to disentangle the impacts of visual experience and gestational age. We however respectfully disagree with the reviewers that an ANCOVA is the appropriate tool for this purpose. The basic assumption behind an ANCOVA is that the covariate factor impacts the dependent variable in a linear fashion. As Figure 2A and 3A show, neither gestational age nor visual experience impact the pattern index linearly; instead, the effects of both factors diminish with age/visual experience. Instead of an ANCOVA, we have therefore included a more basic two-factorial ANOVA with factors age and visual experience. The fact that we find significant main effects of both factors demonstrates that indeed both of them impact the pattern index. We also discuss the dependencies between both factors more directly in the text, which are a consequence of relying on natural eye opening rather than other manipulations of visual experience,

Reviewer #1:The authors addressed most of my concerns. The remaining issues are detailed below.The paper has improved substantially, and the authors have addressed several of my concerns. One substantial issue remains in my opinion. Just like the grouping in developmental epochs felt a bit ad hoc in the previous submission, so does the grouping in visual-experience epochs in this resubmission. It is not clear what makes V4 and V5 so special that these epochs should be at the center of this analysis. I am not sure that the authors have sufficient data to distinguish experience from gestational age. The gestational age difference between the V4 and V5 group is a bit more than a day, and so seemingly highly correlated with the difference in visual experience. Furthermore, I don't understand the logic behind Figure 3D and 3E: Why is the absence of an age-based difference in pattern-selectivity evidence in favor of the role of visual experience? If the authors wish to attempt to isolate the role of visual experience, it would be better to conduct an analysis that involves all of their data, rather than just this subset. For example, a plot like Figure 2A, whereby each animal's median pattern index is shown as a function of visual experience over a reasonable range (e.g., from V1 to V7+ in steps of 1 day) would be more informative. I think that an ANCOVA analysis provides a suitable statistical test for the hypothesis that visual experience explains variance in pattern index beyond the effects of gestational age (Pattern index of each cell/animal: dependent variable; age group: independent variable; visual experience: covariate).

As described above, we now include a figure to show the development of pattern index as a function of visual experience without dividing animals into groups (Figure 3A), and address the impact of age and visual experience on pattern index together in an ANOVA with two factors.